# Structural and functional characterization of NEMO cleavage by SARS-CoV-2 3CLpro

Mikhail A. Hameedi[1,2,3,4,12], Erica T. Prates[4,5,12], Michael R. Garvin[4,5], Irimpan I. Mathews[1], B. Kirtley Amos[6], Omar Demerdash[4,5], Mark Bechthold[3], Mamta Iyer[7], Simin Rahighi[7], Daniel W. Kneller[4,8], Andrey Kovalevsky[4,8], Stephan Irle[9], Van-Quan Vuong[10], Julie C. Mitchell[4,5], Audrey Labbe[4,5], Stephanie Galanie[4,5,11], Soichi Wakatsuki[1,2,3,4] ✉ & Daniel Jacobson[4,5] ✉

In addition to its essential role in viral polyprotein processing, the SARS-CoV-2 3C-like protease (3CLpro) can cleave human immune signaling proteins, like NF-κB Essential Modulator (NEMO) and deregulate the host immune response. Here, in vitro assays show that SARS-CoV-2 3CLpro cleaves NEMO with fine-tuned efficiency. Analysis of the 2.50 Å resolution crystal structure of 3CLpro C145S bound to NEMO$_{226-234}$ reveals subsites that tolerate a range of viral and host substrates through main chain hydrogen bonds while also enforcing specificity using side chain hydrogen bonds and hydrophobic contacts. Machine learning- and physics-based computational methods predict that variation in key binding residues of 3CLpro-NEMO helps explain the high fitness of SARS-CoV-2 in humans. We posit that cleavage of NEMO is an important piece of information to be accounted for, in the pathology of COVID-19.

As of April 2022, Severe Acute Respiratory Syndrome Coronavirus-2 (SARS-CoV-2) has caused over 507 million confirmed cases of COVID-19, more than 6 million deaths (covid19.who.int/), and the global economy to contract by 3.5% in 2020[1]. Unlike previous *betacoronavirus* outbreaks, SARS-CoV-2 has spread to every country, which has provided the urgency and impetus to develop and rapidly distribute therapeutics to reduce the spread of the virus, including RNA-based vaccines. However, many societal impediments, the emergence of variants with enhanced fitness and breakthrough infections have prevented herd immunity from being reached from vaccination coverage, and prove that SARS-CoV-2 eradication is challenging. Therefore, more broadly protective vaccines and effective therapeutic approaches will need to be

implemented to steadily reduce the risk of severe illness and prevent future zoonotic outbreaks.

Current strategies to inhibit viral transmission and to reduce the severity of the disease include disrupting the lifecycle of the pathogen[2,3]. SARS-CoV-2 is an enveloped *betacoronavirus* with a single stranded, positive-sense, 29 kb RNA genome that encodes several open reading frames. *ORF1a* and *ORF1b* encode the polyproteins that are processed to generate the 15 nonstructural proteins of SARS-CoV-2. These include the papain-like protease (PLpro) and the 3C-like protease (3CLpro), which are required to execute the viral life cycle and inhibit the host immune response[4,5]. These proteases therefore represent high-value targets for treatment of COVID19, which is supported by the emergency use authorization granted by the Food and Drug

[1]SLAC National Accelerator Laboratory, Stanford Synchrotron Radiation Lightsource, Structural Molecular Biology, Menlo Park, CA 94025, USA. [2]SLAC National Accelerator Laboratory, Stanford Synchrotron Radiation Lightsource, Biosciences, Menlo Park, CA 94025, USA. [3]Department of Structural Biology, Stanford University, Stanford, CA 94305, USA. [4]National Virtual Biotechnology Laboratory, US Department of Energy, Washington, DC, USA. [5]Biosciences Division, Oak Ridge National Laboratory, Oak Ridge, TN, USA. [6]Department of Horticulture, University of Kentucky, Lexington, KY, USA. [7]Chapman University School of Pharmacy, Irvine, CA 92618, USA. [8]Neutron Scattering Division, Oak Ridge National Laboratory, Oak Ridge, TN, USA. [9]Computational Sciences and Engineering Division, Oak Ridge National Laboratory, Oak Ridge, TN 37831, USA. [10]The Bredesen Center for Interdisciplinary Research and Graduate Education, University of Tennessee Knoxville, Knoxville, TN, USA. [11]Present address: Department of Process Research and Development, Merck & Co., Inc., Rahway, NJ 07065, USA. [12]These authors contributed equally: Mikhail A. Hameedi, Erica T. Prates. ✉e-mail: soichi.wakatsuki@stanford.edu; jacobsonda@ornl.gov

Administration for Paxlovid™ that includes nirmatrelvir, a 3CLpro inhibitor component of the drug.

The Protein Data Bank (RCSB PDB) is replete with structures of wild-type (WT) SARS-CoV-2 3CLpro[6,7]. 3CLpro WT homodimers are a 67.60 kDa, heart-shaped complex[6,8–10]. Each 3CLpro chain consists of three domains. Domain I (aa. 8–101) and Domain II (aa. 102-184) have a predominantly β-sheet structure, form the active site, and contribute to dimerization. Domain III (aa. 201-303) is substantially α-helical and is the primary determinant of dimerization[8,9,11]. The active site of WT 3CLpro contains the catalytic dyad of His41 and Cys145. Prior to substrate-binding, the 3CLpro active site is primed with protonated His41 and a thiolate anion on Cys145. After substrate-binding, the thiolate anion prosecutes nucleophilic attack at the main chain carbonyl carbon of the P1 residue (immediately preceding the substrate scissile bond)[12,13] This leads to heterolytic fission of the scissile bond, followed by active site regeneration.

Functionally, 3CLpro recognizes a hydrophobic substrate residue at P2 (usually Phe or Leu), a Gln at P1, and Ser, Val, Asn, or Ala residues at P1'. This recognition motif is found in multiple sites of the viral polyproteins, which are cleaved by 3CLpro to form mature nsp5-16. This consensus sequence is also present in proteins of the host innate immune pathway and therefore 3CLpro may blunt the immediate antiviral immune response via proteolysis[14]. The NF-κB essential modulator (NEMO)[15] is one of the immune proteins that can be cleaved by 3CLpro[14]. NEMO has been shown to be cleaved by 3CLpro from feline infectious peritonitis virus and porcine epidemic diarrhea virus[16,17]. Recently, N-terminomics experiments also confirmed, in vitro, the cleavage of NEMO by SARS-CoV-2 3CLpro[14].

NEMO is necessary for activating NF-κB during the canonical NF-κB response signaling pathway, which is a critical first response to viral infection. A disrupted NF-κB pathway is a hallmark of chronic inflammatory diseases[18], which suggests that NEMO cleavage by 3CLpro and the downstream dysregulation of NF-κB could contribute to the enhanced inflammatory response observed in COVID-19 patients[15]. Remarkably, consistent with in vitro and in vivo data, Wenzel et al. recently proposed that cleavage of host cell NEMO by 3CLpro is connected to microvascular pathology observed in the brains of COVID-19 patients[19]. Therefore, understanding the molecular basis of NEMO inactivation by 3CLpro can be a platform to develop therapeutic strategies for alleviating symptoms of COVID-19 including pathology in the central nervous system.

The structure of 3CLpro in complex with a NEMO-derived heptapeptide substrate was previously solved for the Porcine Epidemic Diarrhea Virus, an *alphacoronavirus*[20]. Molecular docking and comparative structural analyses suggest that NEMO similarly binds to SARS-CoV-2 3CLpro in a pose that favors proteolysis[21]. However, even a few differences in 3CLpro residues, for example, between SARS-CoV and SARS-CoV-2 3CLpro, were shown to significantly change substrate preferences[14], enforcing the importance of obtaining a high resolution structure of SARS-CoV-2 3CLpro bound to human NEMO and identifying non-conserved interactions. In this work, we first showed that WT SARS-CoV-2 3CLpro can cleave the 33-residue peptide substrate, NEMO$_{215–247}$ using in vitro assays. To explore the molecular basis of this interaction, we solved a 2.50 Å crystal structure of a SARS-CoV-2 3CLpro active site cysteine variant, C145S, in complex with the human decapeptide substrate NEMO$_{226–235}$. This represents the first structure of SARS-CoV-2 3CLpro bound to a human substrate protein. Using this structure as a starting point, extensive molecular dynamics simulations, quantum mechanics calculations, and machine learning-based predictions indicated that the few differences in NEMO and 3CLpro across host species and human-infecting *betacoronaviruses* significantly change the stability of the complex formed between these proteins. Finally, we discuss how ablation of NEMO via proteolysis connects with COVID-19 as a systemic disease.

## Results

### SARS-CoV-2 cleaves NEMO

NEMO is known to form a homodimer of two, 419 residue (49 kDa), mostly α-helical protomers[22,23] and the overall domain architecture is well characterized (Fig. 1a)[23]. Five 3CLpro recognition sites in human NEMO (hNEMO) were identified at Gln83, Gln205, Gln231, Gln304, and Gln313, where the listed Gln would act as the P1 residue in each case[19] (Supplementary Fig. 1). 3CLpro enzyme was incubated with three constructs of NEMO from mouse (mNEMO), *Mus musculus*, with and without a glutathione S-transferase (GST) tag, namely, amino acids 96–250, 221–250, 221–339, GST-96–250, GST-221–250, and GST-221–339, and the products detected by SDS-PAGE were consistent with cleavage following Gln231 (Supplementary Fig. 2). Cleavage at the other potential recognition sites present in these constructs was not detected, suggesting that the Gln231 is more susceptible for proteolysis than the other glutamine sites. To assess the ability and efficiency of 3CLpro to cleave hNEMO, we expressed and purified hNEMO truncated to amino acids 215–247, which mostly contains the Hlx2 domain, where the Gln231 recognition site is located. Concentration-dependent hNEMO$_{215–247}$ cleavage at the site reported for other coronaviruses (aa. 231–232) was detected via high performance liquid chromatography-mass spectrometry (HPLC-MS, Fig. 1b). Multiple sequence alignment showing the identified 3CLpro cleavage site at Gln231 and the 3CLpro cleavage sites in the viral polyprotein is shown (Fig. 1c).

### Coiled-coil propensity analysis of the NEMO dimer

Analysis of coiled-coil parameters of the hNEMO$_{196–251}$ dimer extracted from PDB entry 3CL3 using TWISTER[24] shows an abrupt increase in the coiled-coil local pitch right after residue 241, suggesting a coiled-coil interruption in the region, which may be advantageous for dimer dissociation prior to proteolysis by 3CLpro (Fig. 1d). In other words, this local pitch increase and the resulting increase in interhelical distance lead to an unstable hydrophobic core and a more exposed single helix, which supports the idea that 3CLpro could access the target sequence. The crystal structure used for this analysis does not correspond to NEMO in the ligand-free state but is the coiled-coil part of NEMO bound to the viral ks-vFLIP protein, which might have influenced the TWISTER results. As an additional independent analysis, we used the PAIRCOIL predictor[25], which suggests a lower propensity for coiled-coil formation at the same region, as well as near the other cleavage sites in NEMO. In addition, ANCHOR[26] predicts molecular recognition features (MoRFs) at this region of hNEMO, i.e., in aa. 233–244, respectively (Fig. 1d, e). Based on the cleavage assay experiments and coiled-coil propensity analyses, we decided to use hNEMO$_{226–235}$ peptide for X-ray crystallographic structural analyses.

### Structure of 3CLpro C145S bound to a hNEMO peptide at 2.50Å resolution

After extensive co-crystallization trials, we determined the structure of the SARS-CoV-2 3CLpro C145S variant in complex with a decapeptide from human NEMO at 2.50 Å resolution (Fig. 2a). This structure shows the Michaelis-like complex of SARS-CoV-2 3CLpro C145S and NEMO poised for catalysis.

In the hNEMO peptide-bound 3CLpro C145S structure, the asymmetric unit contains two, mature 3CLpro C145S dimers (Fig. 2a, b). One dimer forms between chains A and B, and the second dimer forms between chains C and D (Fig. 2b). Chains B and C are each bound to the hNEMO peptide (acetyl-KLAQLQVAYH-amide; aa. 226–235) (Fig. 2a, b, c). Within the asymmetric unit, the C-terminal tail (Ser301-Gln306) of chain A inserts into the substrate-binding site of chain D (Fig. 2b, d). This connects the two dimers to each other in the asymmetric unit. Additionally, the substrate-binding site of chain A is bound by the C-terminal tail of chain D from a neighboring symmetry equivalent dimer. This C-terminal tail-binding contributes to crystal lattice formation by connecting chains A and D to each other throughout the

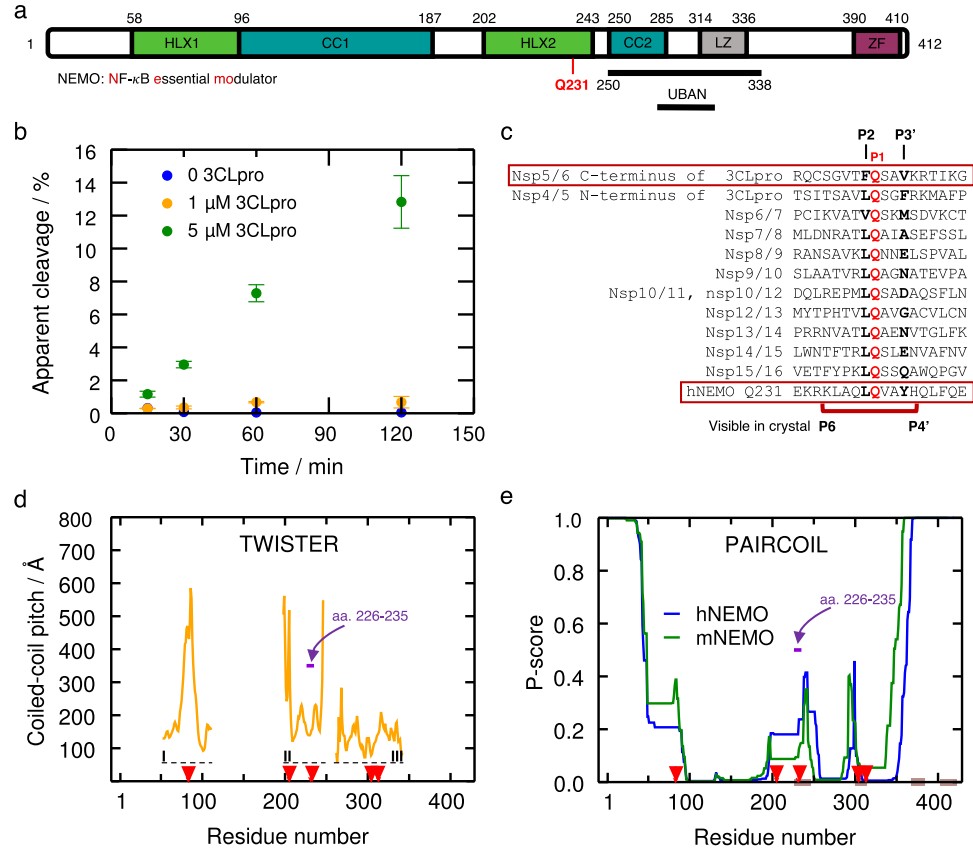

**Fig. 1 | Truncated NF-κB essential modulator (NEMO) is cleaved by 3CLpro in enzymatic assays. a** NEMO includes the α-helical domain 1 (Hlx1), the coiled-coil domain 1 (CC1), the α-helical domain 2 (Hlx2), the coiled-coil domain (CC2), a leucine zipper (LZ) domain, and the C-terminal zinc-finger (ZF). Human NEMO (hNEMO) truncated at site 215 and 247 was used in enzymatic assays with 3CLpro. A recognition site of cleavage is found at Gln231. **b** Cleavage of hNEMO$_{215-247}$ at 0.053 μg/μL (~13 μM) by 3CLpro at two concentrations. Reactions were incubated at 25 °C and aliquots were quenched at different times for analysis. The extent of proteolysis was quantified by LC-MS/MS. Apparent % cleavage was calculated by dividing the product peak area by the sum of the substrate and product peak areas. Error bars represent the range of duplicate enzymatic reactions. Statistics have been derived for $n = 2$ biologically independent experiments. **c** Multiple sequence alignment of peptide sequences of SARS-CoV-2 polyprotein and hNEMO. P1 site

glutamine residues are shown in red. The peptide (P6 to P4') used in the crystal structure is indicated beneath the sequences. **d** Coiled-coil pitch per residue computed with TWISTER[24] for NEMO in the PDB structures 6MI3 (region I)[34], 3CL3 (region II)[33], and 6YEK (region III)[73] is shown. Dashed lines indicate the regions I-III corresponding to these structures. The cleavage sites Gln83, Gln205, Gln 231, Gln304, and Gln313 are indicated with red arrows. The region corresponding to hNEMO$_{226-235}$, used in our X-ray structure determination is pointed out (violet arrow). **e** PAIRCOIL[25] prediction of coiled-coil propensity per residue for human and mouse NEMO (mNEMO). Lower $P$-scores implies greater likelihood of coiled-coil. Potential disordered binding regions predicted by ANCHOR[26] are shown in brown. The cleavage sites are depicted as in **d**. The region corresponding to hNEMO$_{226-235}$ used in our X-ray structure is depicted as in **d**.

crystal (Fig. 2a, b). This was also observed in our crystal structure of 3CLpro C145S without NEMO. Finally, the C-terminal tails of chains B and C bind at sites found at the dimerization interfaces of their respective dimers (Fig. 2e). Similar C-terminal tail contributions to the crystal contacts and the dimerization interface were reported for the C145A mutant of SARS-CoV-2 3CLpro[27]. The hNEMO peptides bind in an extended conformation, similar to an anti-parallel β-sheet with 3CLpro residues Gln189-Ala191 and His163-Pro168. In chains B and C, there is density for residues KLAQLQVAY (aa. 226-234) and, the entire peptide, KLAQLQVAYH (aa. 226-235), respectively (Supplementary Fig. 3a, b), where Lys226 is P6, His235 is P4', Gln231 is P1 and Val232 is P1' (Fig. 2c). His235 may be disordered in the structure, does not have any density and is therefore not modeled in the hNEMO$_{226-234}$ bound into chain B. We have focused on chain B and bound hNEMO$_{226-234}$ rather than chain C and bound hNEMO$_{226-235}$, because chain B has a lower B-factor (33.4 Å²) than chain C (45.6 Å²) and the electron density for hNEMO$_{226-234}$ bound into chain B is higher quality than for hNEMO$_{226-235}$ bound into chain C. The N-acetyl and C-amide caps are not resolved in either hNEMO molecule. The electron density in the substrate-binding sites of chains B and C is unequivocally assigned as the hNEMO peptide, rather than as the C-terminal tail from a

neighboring monomer, since the density on either the N-terminal or C-terminal end of the peptide does not connect to the protein chains near the substrate-binding sites, but projects into the bulk solvent. The peptide density in chains B and C is also too long to be assigned as the C-terminal tail from a monomer in the neighboring dimer, continuing beyond Ser145 in the substrate-binding site (unlike the density for the C-terminal tails in chains A and D). Finally, the positions of all C-terminal tails in the asymmetric unit are accounted for, and the shape of the density in chains B and C around residues P2, P3, P5, and P6 is consistent with the hNEMO sequence. The subsites in the substrate-binding site of 3CLpro are identified as S4 to S4', corresponding to peptide residue positions P4 to P4', where S4 binds to P4 and S4' binds to P4' (Table 1).

**Thr26 and Thr190 of 3CLpro pin the NEMO peptide into 3CLpro**

Thr190 and Thr26 of 3CLpro use hydrogen bonds (H-bonds) to pin the ends of the hNEMO peptide into the substrate-binding site, causing conformational changes in the site. Thr26 and Thr190 form H-bonds with Ala233 (P2') and Ala228 (P4) of the hNEMO peptide, respectively. This causes the distance between the C$_\alpha$ atoms of Thr26 and Thr190 to decrease. This distance decreases from 21.7 Å, in our reported 1.45 Å

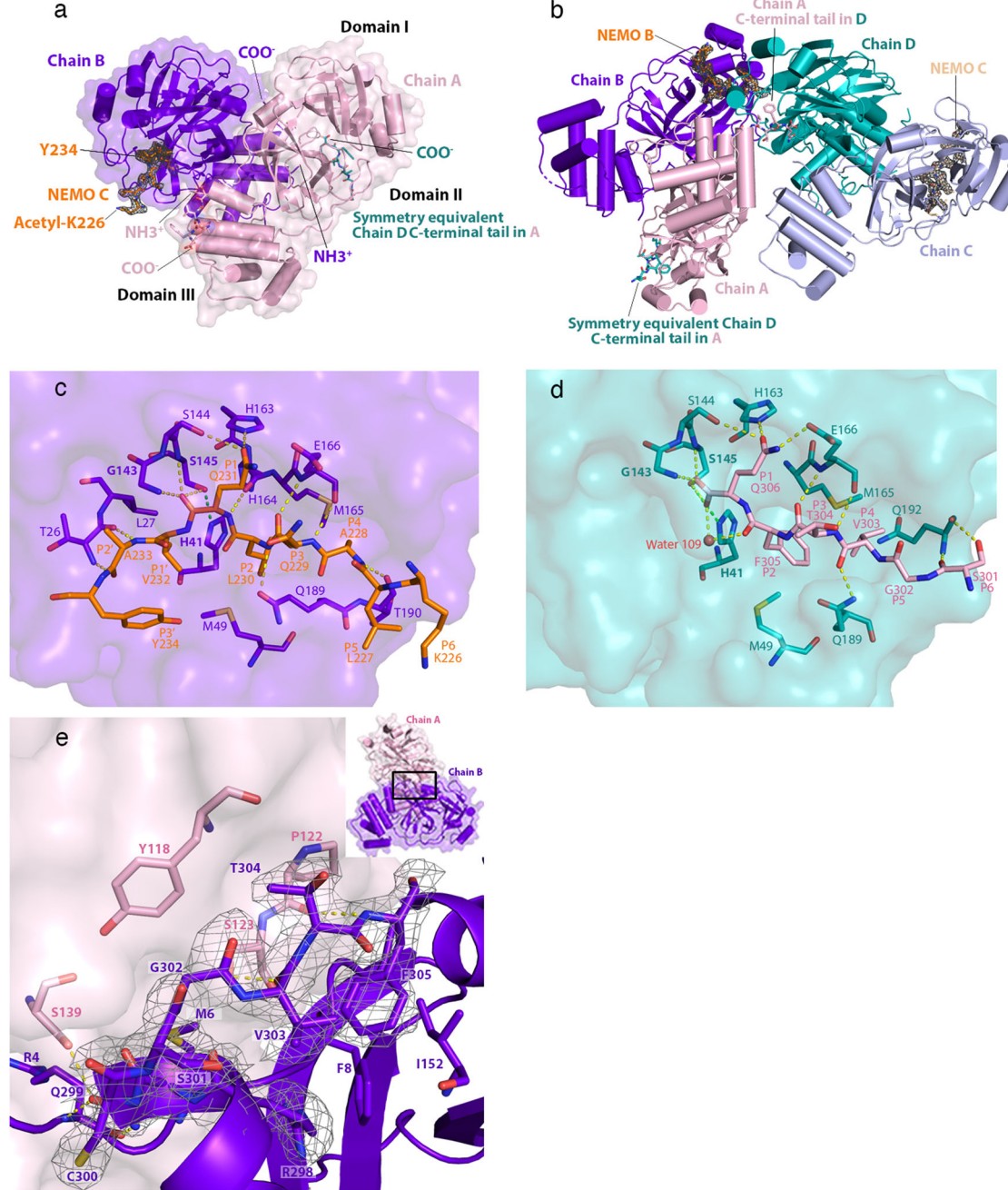

**Fig. 2 | Structure of the hNEMO-bound 3CLpro C145S homodimer. a** The hNEMO$_{226-234}$-bound 3CLpro dimer. N- (NH3 + ) and C-termini (COO-) are labeled. hNEMO$_{226-234}$ (NEMO B - orange) binds into chain B (purple) and is surrounded by an omit map of electron density (1.0 σ contour level and 1.9 Å carving radius). Acetylated Lys226 and Tyr234 at the N- and C-termini of the hNEMO$_{226-234}$ peptide, respectively, are labeled. The C-terminal tail of chain D (teal) from a crystallographic symmetry equivalent binds into the substrate-binding site of chain A (light pink). Domains I, II and III are labeled. **b** Structure of the hNEMO-bound C145S variant asymmetric unit. The C-terminal tail of chain A (pink) binds into the substrate-binding site of chain D (teal). hNEMO peptides bound into chain B (purple) (NEMO B - orange) and chain C (light blue) (NEMO C - wheat) are superimposed with an omit map (1.0 σ contour level and 1.9 Å carving radius) of their electron density. **c** Interactions of hNEMO$_{226-234}$ with the substrate-binding groove of 3CLpro C145S. hNEMO$_{226-234}$ is colored orange. Residues Lys226 to Tyr234 are fully labeled. The surface of chain B (purple) is shown. Substrate-binding residues in

chain B are portrayed as sticks and labeled. Catalytically relevant residues are labeled in bold. Hydrogen bonds are depicted as dashed yellow lines. The hydrogen bond predicted to form between Cys145 (in WT) or Ser145 (in C145S) and His41 is depicted as a dashed green line. **d** Interactions of the C-terminal tail of chain A with the substrate-binding groove of 3CLpro C145S. The C-terminal tail of chain A is colored light pink. Residues Ser301 to Gln306 are fully labeled. The surface of chain D (teal) is shown. Interacting residues and their interactions are portrayed as in **c**. The oxygen atom of water 109 is depicted (red sphere). **e** The C-terminal tail at the 3CLpro dimer interface. The C-terminal tail of chain B (purple) is depicted as sticks, juxtaposed over its cartoon representation. Density from the omit map (1.0 σ contour level and 1.9 Å carving radius) is shown around the C-terminal tail. Arg298 to Phe305 and residues that interact with the C-terminal tail of chain B are labeled. Inset, schematic of chains A and B, showing the position of the C-terminal tail at the 3CLpro dimer interface.

**Table 1 | Interactions of SARS-CoV-2 3CLpro C145S with both bound hNEMO$_{226-234}$ and bound C-terminal tail in the crystal structure**

| Subsite/ Substrate | 3CLpro Residue (moiety) | hNEMO Interaction (distance) | Residue (moiety) | C-terminus Interaction (distance) | Residue (moiety) |
|---|---|---|---|---|---|
| S6/P6 | Q192 (main_N–H) | | | HB (3.0 Å) | S301 (main_C=O) |
| | Q192 (main_C=O) | | | HB (2.9 Å) | S301 (γOH) |
| | N.A. | N.A. | K226 | | |
| S5/P5 | N.A. | N.A. | L227 | N.A. | G302 |
| S4/P4 | T190 (main_C=O) | HB (2.9 Å) | A228 (main_N–H) | | |
| | M165 (side) | | | HC | V303 (side) |
| | Q189 (side_Nε2) | | | HB (2.6 Å) | V303 (main_C=O) |
| S3/P3 | E166 (main_C=O) | HB (3.0 Å) | Q229 (main_N–H) | HB (3.1 Å) | T304 (main_N–H) |
| | E166 (main_N–H) | HB (2.9 Å) | Q229 (main_C=O) | HB (3.1 Å) | T304 (main_C=O) |
| S2/P2 | Q189 (side_Oε1) | HB (3.2 Å) | L230 (main_N–H) | | |
| | M49 (side) | HC | L230 (side) | HC | F305 (side) |
| | M165 (side) | HC | L230 (side) | HC | F305 (side) |
| | Water 109 | | | HB (2.8 Å) | F305 (main_C=O) |
| S1/P1 | H164 (main_C=O) | HB (3.2 Å) | Q231 (main_N–H) | | |
| | S145 (main_N–H) | HB (3.0 Å) | Q231 (main_C=O) | HB (3.0 Å) | Q306 (main_OXT) |
| | G143 (main_N–H) | HB (2.8 Å) | Q231 (main_C=O) | HB (2.7 Å) | Q306 (main_OXT) |
| | S145 (γOH) | HB (3.0 Å) | Q231 (main_C=O) | HB (2.6 Å) | Q306 (main_C=O) |
| | S144 (γOH) | HB (3.0) | Q231 (side_Oε1) | HB (3.2 Å) | Q306 (side_Oε1) |
| | H163 (side_Nε2) | HB (2.4 Å) | Q231 (side_Oε1) | HB (2.5) Å | Q306 (side_Oε2) |
| | Water 109 | | | HB (2.5 Å) | Q306 (main_C=O) |
| | S145 (γOH) | | | HB (3.1 Å) | Q306 (main_OXT) |
| | E166 (side_Oε1) | | | HB (2.6 Å) | Q306 (side_Nε1) |
| S1'/P1' | L27 (side) | HC | V232 (side) | | |
| S2'/P2' | T26 (main_C=O) | HB (3.0 Å) | A233 (main_N–H) | | |
| | T26 (main_N–H) | HB (3.2 Å) | A233 (main_C=O) | | |
| S3'/P3' | N.A. | N.A. | Y234 | | |

The hydrogen bond (HB) interactions involving side chain or main chain atoms and hydrophobic contacts (HC) formed in each subsite of 3CLpro are shown.

resolution WT SARS-CoV-2 3CLpro structure as well as 21.5 and 21.3 Å, in chains B and C respectively, in our reported 2.47 Å resolution 3CLpro C145S structure without NEMO, to 20.5 and 20.3 Å in chains B and C, respectively, in our hNEMO$_{226-235}$ peptide-bound 3CLpro C145S structure (Supplementary Fig. 3c). Comparison to several substrate-free and substrate-bound SARS-CoV-2 3CLpro crystal structures indicates that this conformational change is associated with hydrogen bonds being formed between a substrate and both Thr190 and Thr26 in the S4 and S2' subsites of 3CLpro, respectively (Supplementary Table 1).

### Eight 3CLpro subsites underpin substrate-binding and selectivity

Our structure of 3CLpro C145S bound to a hNEMO$_{226-234}$ peptide identifies key structural sites required for substrate-binding in 3CLpro. The substrate-binding site of 3CLpro consists of a groove made by four groups of consecutive residues, Gln189-Ala191, His163-Pro168, Phe140-Ser145, and Thr24-Leu27, as well as His41 and Met49. Gln189-Ala191 and His163-Pro168 run adjacent to each other, bind the P2-P4 residues of the hNEMO peptide and position P1 in the active site. In the active site, Phe140-Ser145 and His163 form a tight S1 subsite for binding P1. Following the active site, Thr24-Leu27 binds the P1'-P3' residues. His41 is involved in catalysis and Met49 interacts with P2.

Table 1 indicates that the preference for main chain H-bond interactions with the substrate retains substrate versatility in the S4-S2 and S1'-S3' subsites. This supports the versatility of 3CLpro in cleaving the viral polyprotein at multiple sites and its activity towards several host proteins (Fig. 1c)[14,15]. There are no S5 and S6 subsites in our NEMO-bound 3CLpro structure for P5 and P6 substrate residues, respectively.

To modulate substrate-selectivity, hydrophobic contacts select for hydrophobic side chains at P2, and occasionally at P4 and P1'. Furthermore, in the S1 subsite, main chain interactions from His164, as well as Gly143 and Ser145 of the oxyanion hole, combine with side chain interactions from His163 to select for glutamine at P1. These interactions position the main chain carbonyl carbon of Gln231 for nucleophilic attack during catalysis and stabilize the tetrahedral anion intermediate during heterolytic fission. The side chain of Ser145 forms H-bonds with the main chain carbonyl of Gln231.

### C-terminal tail versus hNEMO-binding in the substrate-binding site

Comparison of our hNEMO$_{226-234}$ and C-terminal tail-bound structures identifies that the S1, S2, S4 and S6 subsites contribute to substrate versatility in 3CLpro and that the S3 subsite interactions with P3 residues are conserved between substrates. Unlike the C-terminal tail, there are no interactions between 3CLpro and the P6 residue of hNEMO$_{226-234}$. Furthermore, hNEMO$_{226-234}$ forms a H-bond with Thr190 in the S4 subsite (Table 1), while in the C-terminal tail, the side chains of Gln189 and Met165 respectively form a H-bond and hydrophobic contact in the S4 subsite. Additionally, in the S2 subsite, hNEMO$_{226-234}$ forms a H-bond with Gln189, compared to the C-terminal tail, which forms a H-bond with water 109. The S4 and S2 subsites therefore engage different interactions to bind hNEMO$_{226-234}$ P4 and P2 residues respectively while S6 does not

interact with hNEMO$_{226-234}$ at all. Finally, C-terminal P2 (Phe305) and P1 (Gln306) form H-bonds with water 109. This water is deacylating during 3CLpro catalysis[27], 3.1 Å from the C-terminal carboxyl carbon and has a Bürgi-Donitz angle of 132° (Fig. 2d). In the hNEMO peptide-bound structure, it is displaced by a hydrophobic side chain of P1'. Our C145S construct terminates at the P1 position (Gln306) and so lacks the residues corresponding to P1' to P3'. The structure of 3CLpro bound to a longer C-terminal tail with residues P1' to P3' would help identify interactions in the S1'-S3' subsites of 3CLpro that contribute to substrate-binding versatility.

## C-terminal tails bind at two distinct sites of the 3CLpro dimer

An alternative binding site for the C-terminal tail is found in a groove formed at the dimerization interface (Fig. 2e). Here, the C-terminal tail is sandwiched between the two β-strands from Gly109-Tyr118 and Ser121-Arg131 in the neighboring chain in the dimer and both the Ser1-Gly11 loop and the Asp153-Cys156 turn in the same chain. Specifically, in chain B, a C-terminal α-helix positions residues 300-305 at the dimer interface via H-bonds. The side chain methyl group of Thr304 forms hydrophobic contacts with the side chain of Tyr118 in chain A. Additionally, the side chain of Phe305 forms hydrophobic contacts with a hydrophobic pocket consisting of the side chains of Phe8, Ile152 and Val303 in chain B. Residues in chain A also form H-bonds with C-terminal residues in chain B. Specifically, the main chain amide of Phe305 forms a H-bond with the main chain carbonyl of Pro122 and the γOH groups of Ser123 and Ser139 form H-bonds with the main chain carbonyl of Val303 and Gln299 Oε1, respectively. Finally, Gln299 Oε1 and Nε2 respectively form H-bonds with the main chain amide and carbonyl of Arg4 in chain B. It is possible that this interfacial site has a role in positioning substrates and cleavage products prior to and following catalysis, respectively.

## Interfacial C-terminus may attenuate self-inhibition in 3CLpro dimer

Estimation of the binding affinity by rigid re-docking of a C-terminus-like peptide (Cys300-Gln306) to the two alternative C-terminal binding sites predicts stronger binding to the catalytic site compared to the dimer interface, with AutoDock Vina binding energies of −14.5 vs. −12.8 kcal/mol, respectively. The good ranking power of AutoDock Vina[28] and the presence of both states in our crystallographic structure of SARS-CoV-2 3CLpro suggest comparable binding affinities in vitro between the two binding sites. Therefore, the conformational change of the C-terminal tail between alternative binding sites is a plausible hypothetical mechanism of modulating 3CLpro inhibition by the C-terminal tail upon homodimerization. In line with that and with the hNEMO-cleavage detected in vitro, the interaction with the C-terminal tail at the active groove is predicted to be less effective compared to hNEMO$_{227-234}$-binding (−19.8 kcal/mol).

## Molecular dynamics simulations of the WT 3CLpro bound to the NEMO peptide

Five independent runs of the WT 3CLpro dimer (chains A and B, carrying catalytic Cys145) bound to hNEMO$_{227-234}$, using our crystal structure as the initial configuration, show that the H-bonds identified in the static structure persist (Supplementary Table 2, Fig. 3a). Additional H-bonds are formed between the residue pairs Ala228-Gln189, Gln231-Cys145, and Val232-Asn142, in hNEMO$_{227-234}$ and 3CLpro, respectively, appearing during about 20-30% of the simulation time. The H-bond pair Leu230-Gln189 appears less than 7% of the time. Several other residues in the binding site of 3CLpro form stable contacts with hNEMO$_{227-234}$ (Fig. 3a). Finally, using the crystal structure as input (chain B, NEMO B), the machine learning (ML)-based predictor, KFC2[29,30] indicates that hot spots, i.e., residues likely accounting for most of the binding affinity, coincide with those forming persistent contacts between 3CLpro and hNEMO$_{227-234}$ (Fig. 3a).

## Predicted NEMO binding affinity differs among host and virus species

The binding core of NEMO is highly conserved among different species (Fig. 3b), but amongst mammals, the predicted hot spot at P1' is a Val232 in hNEMO whereas it is an Ala232 in mNEMO and golden hamster (*Mesocricetus auratus*) NEMO. In order to understand the relative contributions of the P1' residues in human and mouse, we then carried out other MD simulations for hNEMO$_{227-234}$ and mNEMO$_{227-234}$. Like in hNEMO, the distance between the catalytic S$^-$ in 3CLpro Cys145 and the carbonyl C in mNEMO Gln231 remains around 5.0 Å (Fig. 3c). However, the simulations show a decrease in the average number of contacts between mNEMO ($49 \pm 4$) and 3CLpro compared to hNEMO ($54 \pm 4$), and the C$_\alpha$ root-mean-square deviation (RMSD) of mNEMO ($1.9 \pm 0.6$ Å) relative to the initial position is larger than that of hNEMO ($1.4 \pm 0.4$ Å). We hypothesize that the impact of V232A may become more pronounced with a longer construct of NEMO in the dimeric form, as competing interactions within the NEMO dimer may more effectively destabilize mNEMO than hNEMO, which appears to be more strongly bound to the catalytic site of 3CLpro. NEMO dimerizes at the Hlx2 region forming hydrophobic contacts between the pairs of Tyr234, Leu230, and Leu227 in each protomer[22], and Tyr234 forms H-bonds with Glu240 in the neighboring protomer. In agreement, PAIRCOIL predicts higher coiled-coil propensity near the Gln231 cleavage site for mNEMO compared to hNEMO (Fig. 1e), which could be a result of the higher helix-formation propensity of alanine compared to valine.

The viral counterpart, 3CLpro, is also highly conserved among species, but structural differences between human-infecting *betacoronaviruses* underpin differences in their predicted interactions with NEMO. Specifically, in HCoV-HKU1, MERS-CoV, SARS-CoV, and SARS-CoV-2, the predicted hot spots, Met49, His164, and Met165, are not fully conserved, and the loop/α-helix formed by aa. 41-53 in Domain I of 3CLpro, which skirts the substrate-binding site, harbors most of the other differences. MD simulations of these proteins bound to hNEMO$_{227-234}$ show that 3CLpro from MERS-CoV ($43 \pm 4$) and SARS-CoV ($46 \pm 7$) exhibit a lower average number of contacts than SARS-CoV-2 ($52 \pm 3$) and HCoV-HKU1 ($53 \pm 2$) (Fig. 4a–d). Particularly, this corroborates the hypothesis, detailed in Discussion, that the S46A substitution between SARS-CoV-2 and SARS-CoV 3CLpro significantly impacts interactions with hNEMO$_{227-234}$. Unsurprisingly, given the proximity of S/A46, the hot spot Met49 is one of the affected contacts. Simulations of these enzymes in the ligand-free state show that the aa. 41-53 in 3CLpro exhibit slightly different flexibility, with SARS-CoV 3CLpro displaying root-mean squared fluctuations of 2.2 and 2.3 Å in each chain of the dimer and SARS-CoV-2 3CLpro, 2.3 and 2.5 Å. Additionally, essential C$_\alpha$ cross-correlation analysis suggests that SARS-CoV-2 3CLpro dimer has significantly more residue pairs exhibiting synchronized motions along the same or opposite directions than SARS-CoV 3CLpro. This may reflect a tighter dimeric packing (38,183 vs. 22,513 pairs and 35,707 vs. 17,567 pairs, respectively; cutoff is a correlation modulus of 0.85; Supplementary Fig. 4). More synchronized pairs are also found when the starting configuration is our originally ligand-bound structure of SARS-CoV-2 3CLpro (chains A and B) with the NEMO peptide excluded, i.e., 37,128 and 29,955 positively and negatively correlated pairs, respectively.

Finally, for a quantitative assessment of relative hNEMO-binding affinities, quantum mechanics (QM)- and molecular dynamics/machine learning (MD/ML)-based predictions were carried out. In the QM calculations, the fragment molecular orbital density-functional tight-binding (FMO-DFTB) method[31,32] was used. For the MD/ML-based approach, five different models were trained. The rankings from the weakest to the strongest binding enzyme were nearly consistent between the two approaches (Fig. 4e and Supplementary Table 3). The exception is the relative position of SARS-CoV 3CLpro swapped with HCoV-HKU1 in the ranking of the FMO-DFTB method. A possible

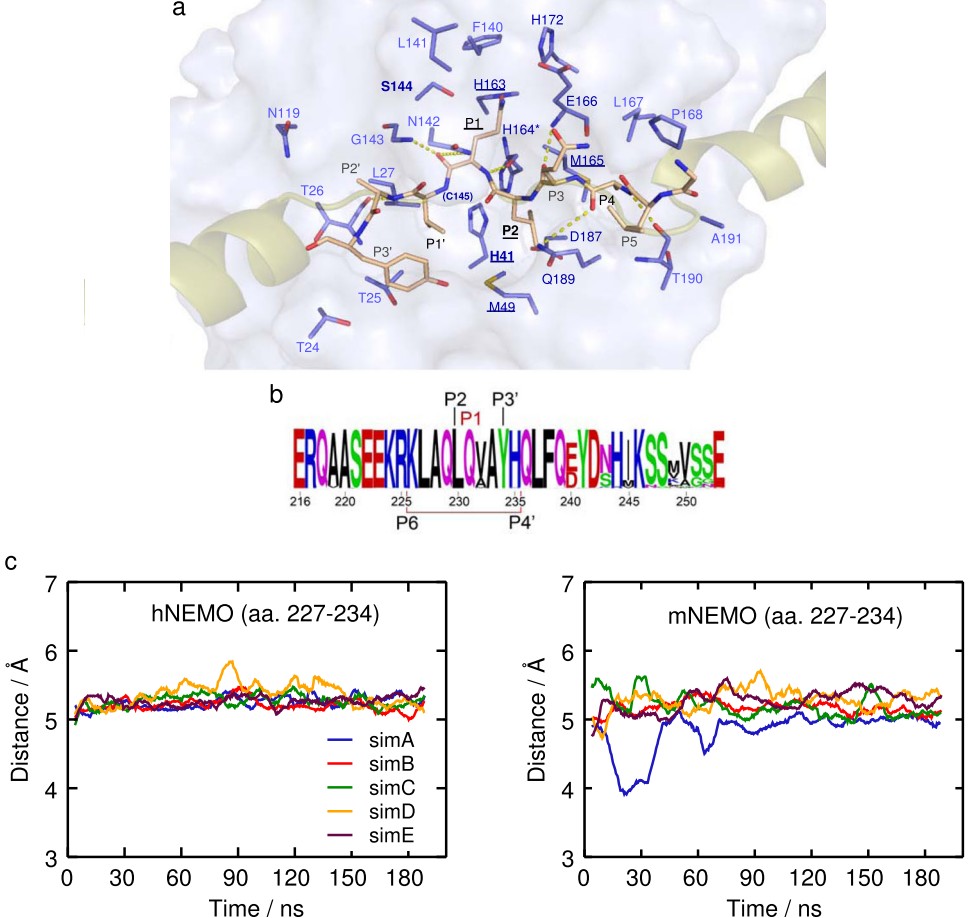

**Fig. 3 | Molecular dynamics simulations of SARS-CoV-2 3CLpro bound to human or mouse NEMO. a** Main contacts from MD simulations and predicted hot spots in WT 3CLpro bound to human NEMO$_{227-234}$. Contacts that persist for more than 70% of the simulation time are depicted. Hot spots predicted with either KFCa or KFCb are labeled in bold and those predicted with both methods are underlined. Persistent H-bonds are shown as dashed lines (Supplementary Table 2). A representative conformation of the short (aa. 227-234) and long constructs of NEMO in the binding site is shown as solid and transparent surfaces, respectively.
**b** Sequence Logo generated with WebLogo[74] of NEMO (aa. 216-253) across 535 animal sequences, including those from placentals, bats, marsupials, birds, rodents, and primates. Clustal Omega[75] was used for the multiple sequence alignment and as an input for the WebLogo analysis. **c** Time evolution of the distance between the catalytic S$^-$ in 3CLpro Cys145 and the carbonyl C of Gln231 in hNEMO$_{227-234}$ and mNEMO$_{227-234}$ computed from MD simulations.

explanation, further explored in Discussion, is that the few differences between SARS-CoV and SARS-CoV-2 3CLpro affect the conformational flexibility of the binding site, which may be captured in the MD/ML but not in the QM approach.

It is not fully clear to which extent the use of a crystal structure only for the hNEMO-peptide bound SARS-CoV-2 3CLpro impacts these predictions. However, we find evidence that suggests the robustness of our predictions and encourage future binding assays to evaluate them. For example, the highest predicted binding affinity of the NEMO peptide to SARS-CoV-2 3CLpro could be thought of as a biased result from using a crystal structure as the starting point for this case. However, we find that energy minimized-only structures, which are much less perturbed than those used in the MD/ML protocol, do not provide consistent results like our methods and do not confirm this bias, with SARS-CoV-2 3CLpro presenting lower relative binding affinity to the hNEMO peptide compared to other coronaviruses' 3CLpro (Supplementary Table 4).

## Discussion

SARS-CoV-2 3CLpro binds Lys226-Tyr234 of the α-helical Hlx2 domain of one hNEMO protomer in an extended conformation. In agreement, the QM calculations predict favorable energetics for hNEMO-binding to 3CLpro in such an extended conformation. SARS-CoV-2 3CLpro

therefore either makes use of a transient partially unwound state of Hlx2 for proteolysis or it actively outcompetes hNEMO dimer interactions and unwinds the α-helix of one protomer. Specifically, 3CLpro forms two hydrophobic contacts as well as H-bonds with Leu230. This outcompetes the hydrophobic contacts formed by Leu230 in the hNEMO dimer partner. Additionally, 3CLpro forms H-bonds with hNEMO at Ala233, next to Tyr234, outcompeting the H-bond formed between Tyr234 and Glu240 in the hNEMO dimer.

ANCHOR-predicted disorder-to-order transition upon binding (Fig. 1d) seems consistent with the extended conformation of the NEMO peptide bound to 3CLpro observed in our crystal structure (Fig. 2c). We speculate that the overlap between the relatively low propensity to form coiled-coil structure and the predicted MoRF may partially explain a higher susceptibility to proteolysis at site Gln231 compared to other cleavage sites (Supplementary Fig. 2). However, the precision of these analyses are limited to the use of predictors and to NEMO structures that may not accurately represent the state preceding its binding to 3CLpro, i.e., in PDB ID 3CL3[33], NEMO interacts with other proteins and in PDB ID 6MI3[34] it is fused to N- and C-terminal ideal coiled-coil adapters.

NEMO homodimers are proposed to form three cellular structures, i.e., homodimers, higher-order lattices, and signalosome-proximal aggregates[35]. The higher-order lattice forms when the

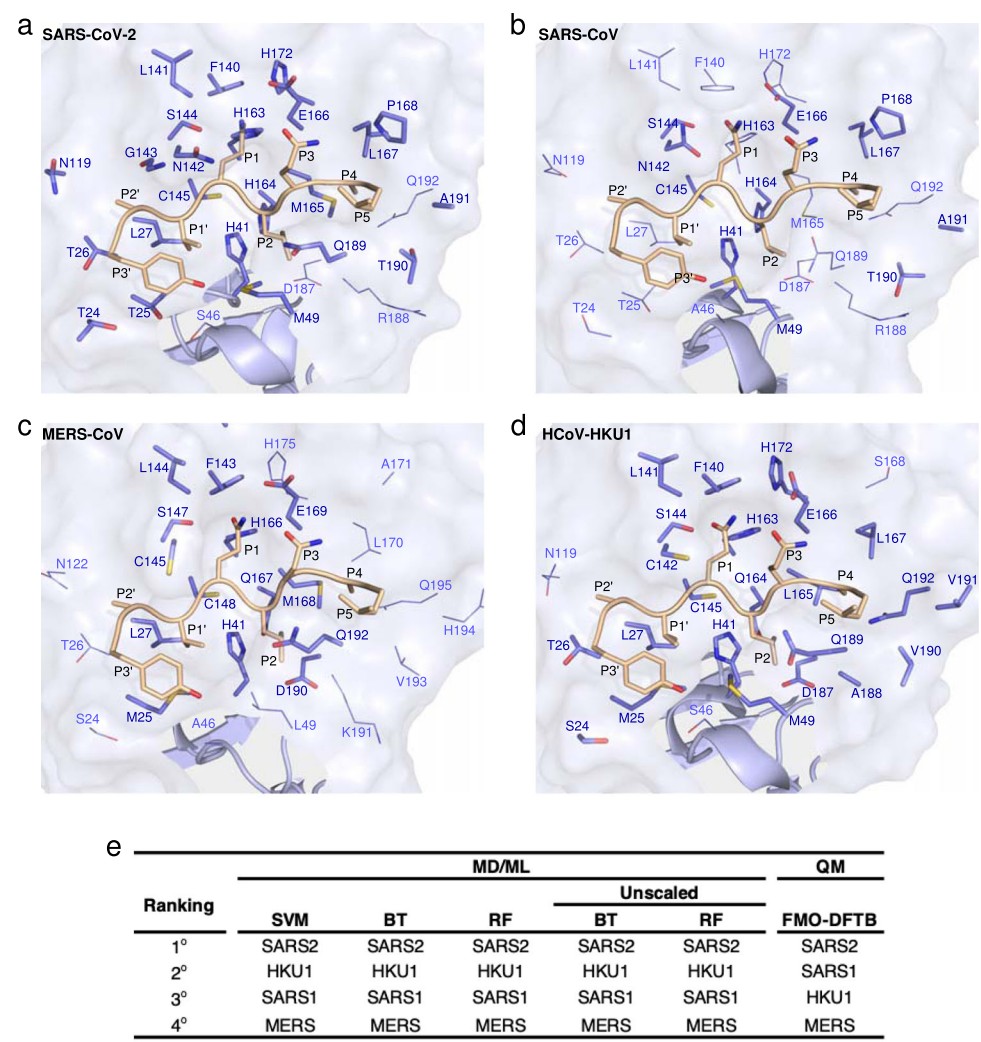

**Fig. 4 | Comparative analysis of 3CLpro from human-infecting *betacoronaviruses* bound to NEMO. a–d** Substrate-binding site of human-infecting *betacoronaviruses* with hNEMO$_{227–234}$. Contacts that persist for more than 70% of the simulation time are labeled in bold. **e** Ranking of predicted hNEMO$_{227–234}$-binding affinities to 3CLpro from *betacoronaviruses* computed with quantum mechanics (QM) and molecular dynamics/machine learning (MD/ML) approaches. SARS-CoV-2, SARS-CoV, HCoV-HKU1, and MERS-CoV are abbreviated as SARS2, SARS1, HKU1, and MERS, respectively. In the MD/ML approach, five machine learning methods were used to train the model, namely, support-vector machine (SVM), gradient-

boosted trees (BT; scaled and unscaled*), and random forest (RF; scaled and unscaled*). The ranking displayed was consistent for nine of the five cases using MD conformers (five ML models applied to either all MD conformers or the three lowest-energy MD conformers). The exception was a boosted tree model trained on unnormalized features that yielded a ranking of SARS-CoV < MERS-CoV < HKU1-CoV < SARS-CoV-2 when considering just the conformers with the lowest energy of interaction with 3CLpro computed from MD simulations. *Unscaled refers to the fact that unscaled, or unnormalized, features were used in the training.

ubiquitin binding domain of one NEMO homodimer binds a linear-ubiquitin chain on another homodimer and two IKKα/β domains form a head-to-head hetero tetramer, thus bringing two sets of IKKα/β-NEMO complexes together[35]. These NEMO lattices compact to form the signalosome-proximal aggregates during NF-κB signaling stimulation by interleukin-1 (IL-1)[35]. Higher-order structures cooperatively enhance NF-κB signaling. Cleavage at Gln231 in the hNEMO homodimer would separate the NEMO kinase-binding site from the ubiquitylation and ubiquitin-binding sites, preventing IKKα/β activation, IKK assembly, and thus lattice formation, which all ultimately ablate the NF-κB signaling. It is unknown if and to what extent Gln231 is accessible to 3CLpro in the hNEMO aggregates. Future characterization of the different structural levels of NEMO as well as how they interchange in equilibrium would help to answer this question.

Comparison of our hNEMO peptide-bound structure solved under cryo-conditions with the room-temperature crystal structure of SARS-CoV-2 3CLpro C145A bound to a peptide of its N-terminal

autoprocessing sequence, acetyl-SAVLQSGF-amide (PDB ID 7N89)[36] identifies specific interactions engaged by 3CLpro to bind either the N-terminal or hNEMO substrate, as well as conserved interactions between the two structures (Fig. 5a). Regardless of differences in interactions between the two structures, the overall binding poses of N-terminal and hNEMO$_{226–234}$ substrates are conserved (Fig. 5b).

In 7N89, the S5 subsite recruits a water molecule and main chain H-bonds from Pro252 and Gly251 to bind N-terminal P5 (Ser). This is not observed in our hNEMO$_{226–234}$-bound structure. The S4 subsites in both hNEMO- and N-terminal-bound structures share conserved enzyme–substrate interactions. In the S3 subsite, the main chain H-bonds formed with Glu166 are conserved when binding both substrates. However, 3CLpro recruits an additional hydrophobic contact between the alkyl moiety of Glu166 and N-terminal P3 (Val). The S2 subsite conserves all interactions observed between 3CLpro and the hNEMO peptide but engages an additional main chain H-bond with water476 that stabilizes N-terminal P2 (Leu) in its subsite. 7N89

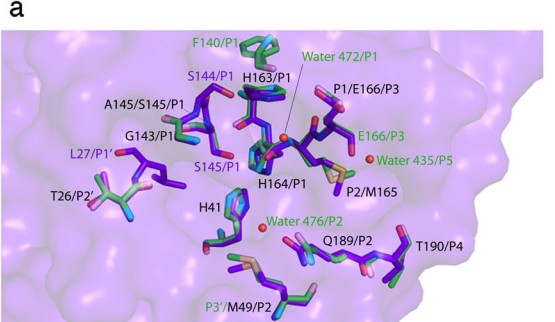

**Fig. 5 | Comparison of 3CLpro interactions with its N-terminal sequence and the hNEMO$_{226-234}$ peptide. a** The surface of chain B from our hNEMO$_{226-234}$-bound structure is depicted as in Fig. 2c. This is juxtaposed with residues from our structure that form interactions with the hNEMO$_{226-234}$ peptide (purple sticks) as well as N-terminal interacting residues (green sticks) from a structure of 3CLpro bound to an N-terminal peptide (PDB_ID:7N89). For 7N89, oxygens are colored light red, nitrogens are colored aquamarine and sulfurs are colored light yellow, for ease of comparison. Residues forming conserved interactions are labeled in black. Those found only in hNEMO$_{226-234}$-bound 3CLpro are labeled in purple. Those found only in 7N89 are labeled in green. Water molecules are portrayed as red spheres and are exclusive to 7N89. Leu27 forms a hydrophobic contact with P1' of hNEMO$_{226-234}$ only. Met49 forms a hydrophobic contact with the P2 residues of both hNEMO$_{226-234}$ and the N-terminal peptide, but also forms a hydrophobic contact with P3' of the N-terminal sequence. **b** Comparison of bound hNEMO$_{226-234}$ from our structure with bound N-terminal peptide from 7N89 shows a conserved substrate pose. hNEMO$_{226-234}$ is colored as in Fig. 2c. The N-terminal peptide carbons in 7N89 are colored dark brown and the oxygens and nitrogens are colored as in panel **a**. Substrate residues are labeled P6-P3'.

identifies extra interactions that strengthen the specificity for P1 (Gln). In 7N89, the side chain of P1 (Gln) forms an additional H-bond with water472 as well as the side chain of Glu166 and the main chain of Phe140. The Ser144 γ-hydroxyl group in 3CLpro C145S forms an additional H-bond with the carbonyl group of Gln P1 in the hNEMO$_{226-234}$-bound structure. The S1' subsite is flexible. It forms no interactions with the N-terminal substrate but engages a hydrophobic contact from the side chain of Leu27 to bind hNEMO$_{226-234}$. The S2' subsite interactions are conserved between the hNEMO peptide- and N-terminal-bound structures, indicating a requirement for Thr26 peptide–nonspecific main chain interactions at this subsite. Finally, 3CLpro engages a hydrophobic contact between Met49 and Thr24 to bind N-terminal P3' (Phe) but not hNEMO$_{226-234}$ P3'.

A broader comparison with peptide-bound 3CLpro structures (Supplementary Table 5) reveals substrate-binding groove plasticity underlying the substrate versatility of this enzyme. This plasticity encompasses the addition or removal of a few H-bonds and hydrophobic interactions, entire subsites and, finally, interactions with structural water molecules. For example, in the structure of SARS-CoV-2 3CLpro C145A bound to the C-terminal autoprocessing sequence (PDB ID 7JOY)[27], not only are several different specific interactions formed, but an additional subsite, S6, is also defined. In turn, this comparison also reveals that many interactions are conserved between different substrates. Specifically, the interactions in S3 are highly conserved, even in the structure of the SARS-CoV 3CLpro H41A mutant bound to the N-terminal autoprocessing sequence (PDB ID 2Q6G)[37]. This conservation of main chain H-bonds in the S3 subsite highlights the importance of the S3 subsite in versatile substrate-binding by 3CLpro.

Subtle structural differences between otherwise highly conserved *betacoronavirus* 3CLpro enzymes are predicted to have substantial effects on NEMO-binding. Our computational analyses suggest that the substitution at residue 46 in the substrate-binding site from serine to alanine in SARS-CoV-2 and SARS-CoV 3CLpro, respectively, increases local rigidity. Small differences in interaction profiles and molecular rigidity observed in MD simulations of intrinsically disordered proteins were demonstrated to predict significant differences in proteolytic efficiency[38]. Similarly, we hypothesize that this substitution in 3CLpro impairs conformational changes that optimize the interactions with NEMO[38]. Alanine has significantly higher α-helical propensity than serine[39] and, indeed, Ala46 is part of an α-helix while Ser46 is at a turn in ligand-free SARS-CoV[40] and SARS-CoV-2 3CLpro structures,

respectively. MERS-CoV 3CLpro has Ala46 and Pro45, both known to increase local rigidity[41], and HCoV-HKU1 lacks such rigidifying residues. This is consistent with the predicted decrease in NEMO affinity from SARS-CoV-2 through HCoV-HKU1, SARS-CoV, and MERS-CoV. This may also explain why an N-terminomics analysis identified more substrates for SARS-CoV-2 3CLpro than for SARS-CoV 3CLpro despite 96% identity[14] and why FRET experiments did not capture any activity of SARS-CoV 3CLpro on NEMO[17].

Differences distal to the binding site may also have a significant functional impact. For example, the presence of Thr285 in the Domain III of SARS-CoV 3CLpro is associated with a looser dimer and slightly lower catalytic efficiency compared to SARS-CoV-2 3CLpro, which has Ala285[10]. Essential C$_\alpha$ cross-correlation analysis of molecular dynamics simulations shows that this difference in dimerization was captured in our trajectories. Although a causal relationship is not obvious, our results reinforce the importance of accounting for molecular differences in the entire proteome for a complete understanding of pathogenesis.

Similarly, we predict that the substitution V232A in NEMO from human to mouse affects NEMO-binding to 3CLpro. Intriguingly, in contrast to the low cleavage efficiency in the HPLC-MS assay with hNEMO$_{215-247}$ (Fig. 1), we found a relatively fast reaction completion of 20 min with mNEMO$_{221-250}$ in the PAGE assay (Supplementary Fig. 2). However, systematic cleavage assays, i.e., using the same experimental conditions, are essential to determine the change in cleavage efficiency. Residue 232 is variable in other relevant host species of SARS-CoV-2 and may have implications in the disease tolerance (Supplementary Table 6)[42]. Therefore, animal models incorporating multiple elements of the human immune system, as in mice transplanted with human immune cells[43], may be a way of accounting for the impact of apparently subtle differences in pathogenesis.

The interaction of 3CLpro with host proteins that are part of the innate immune response appears to be finely tuned to avoid competition with its role of processing the viral polyproteins and to maintain functional cells for production of virions[44]. Wenzel et al. determined an apparent catalytic efficiency at Gln231 in a hNEMO peptide by 3CLpro of about 43 s$^{-1}$ M$^{-1}$ and pointed out that this is in the range reported for the cleavage between nsp4 and nsp5[19,45]. This may not be simply extrapolated to the cell environment, where other factors can significantly influence reaction rates (e.g., attenuated diffusion rates)[46]. Our enzymatic assays indicate that cleavage of NEMO is a slow process and likely not impactful in the first hours of infection[47]. Similarly, it was

demonstrated that SARS-CoV-2 infection of different cell lines requires 24–48 h to cause a dramatic reduction in the levels of host proteins cleaved by PLpro and 3CLpro[15]. These observations and the cleavage of multiple host proteins detected with N-terminomics suggest that the cleavage of NEMO is one component of a number of mechanisms that SARS-CoV-2 combines to counteract the host immune response[14]. However, certain traits of COVID-19 are remarkably consistent with known effects caused by NEMO ablation, suggesting its particular relevance in the pathophysiology. Mutations in the *NEMO* gene in the genetic disease *incontinentia pigmenti* are typically lethal in males and cause skewed X-inactivation patterns in females, selecting for the normal allele, as *NEMO* resides on the X-chromosome[48]. Similarly, males seem to develop more severe COVID-19[49]. Additionally, deletion of *NEMO* leads to rarefaction of brain microvessels[50] and increased vascular permeability in the brain is observed in COVID-19 patients with neurological symptoms[51], which resembles those of *incontinentia pigmenti*[52]. Finally, evidence of the biological relevance of this study is provided in Wenzel et al., which demonstrated that, by cleaving NEMO, 3CLpro induces the death of human brain endothelial cells[19].

NEMO lies at the nexus of the antiviral response driven by the mitochondrial antiviral-signaling protein (MAVS) that results in the activation of both NF-κB and type I interferons. The NF-κB pathway is targeted by diverse viruses[53] and its suppression contributes to an imbalance in the renin-angiotensin system, which is proposed to result in severe COVID-19 outcomes[54]. Our results suggest that SARS-CoV-2 weakens innate immunity by cleaving NEMO using 3CLpro. In addition to directly countering the immunosuppressive effects from NEMO cleavage, inactivating 3CLpro as a therapeutic strategy would impair viral replication and reduce the production of proteins that intervene downstream from MAVS, such as nsp6, and nsp13[55].

Future avenues for research will involve defining the link between the binding affinity of 3CLpro to NEMO and the observed proteolytic rate via enzymatic assays. We expect that our results motivate site-directed mutagenesis experiments to quantify the contribution to proteolytic efficiency of the few non-conserved amino acids in 3CLpro and NEMO across virus and host species, respectively. Finally, semi-quantitative analysis of accumulation of 3CLpro and the products of NEMO-cleavage in human cells from different tissues infected with SARS-CoV-2 will help to unravel the specific pathogenesis traits derived from ablation of NEMO.

## Methods

### NEMO peptide expression and purification
The constructs of Human NEMO (residues 215-247) and mouse NEMO (residues 221-250) cloned into a pGEX-6p-1 vector were transformed into BL21 (DE3) cells and selected using ampicillin-enriched LB media. Cells were grown to OD600 of 0.6-0.8 and induced with 0.25 mM IPTG overnight at 25 °C. Cells were harvested, resuspended in phosphate buffered saline (PBS) and lysed on ice using a sonicator (QSonica). Insoluble material was pelleted using centrifugation and the lysate was incubated with Glutathione Sepharose 4B (GS4B, Cytiva) for 2 h on a rotating platform at 4 °C. The beads with the GST-tagged protein were separated using a gravity column and washed with PBS to remove any unbound protein. The GST tag was cleaved on-column using Prescission protease and incubating overnight at 4 °C. Cleavage was confirmed by SDS-PAGE and the cleaved protein was eluted using PBS. The protein was further purified by size-exclusion chromatography on a Superdex 75 16/60 column using 20 mM Tris-HCl, pH 8.0 and 150 mM NaCl as the running buffer.

### 3CLpro expression and purification
3CLpro WT enzyme for assays was prepared independently from a clone of the SARS-CoV-2 NSP5 gene in pD451-SR (Atum, Newark, CA) according to published procedure[8]. To make the authentic N-terminus, the protease sequence is preceded by maltose binding protein followed by the 3 CLpro autoprocessing site between nsp4 and nsp5 (SAVLQ ↓ SGFRK, arrow indicates the cleavage site). Authentic C-terminus is achieved by a sequence of a human rhinovirus 3C (HRV-3C) cleavage site (SGVTFQ ↓ GP) connected to a His6 tag. The N-terminal flanking sequence is autoprocessed during expression in *E. coli* (BL21 DE3), whereas the C-terminal flanking sequence is removed by the treatment with HRV-3C protease (Millipore Sigma, St. Louis, MO) in-between two rounds of Ni-NTA affinity chromatography.

### Human NEMO cleavage assay
70 μL reactions of substrate hNEMO 215-247 (0.053 μg/μL) and WT 3CLpro (0, 1, or 5 μM) were prepared in duplicate using a reaction buffer of 20 mM Tris-HCl, pH 7.35, 100 mM sodium chloride, 1 mM EDTA, and reduced 2 mM glutathione. Reactions were incubated at 25 °C with gentle shaking and 5 μL aliquots were quenched by diluting into 95 μL of 1.63% formic acid in water at 4 °C at 15, 30, 60, and 120 min. 2 μL of quenched reactions were injected onto an Agilent EclipsePlusC18 1.8 μM, 2.1 × 50 mm chromatography column, and eluted using a gradient elution of 2–80% buffer B (0.1% formic acid in acetonitrile) against buffer A (0.1% formic acid in water) over 8.5 mins. Samples were introduced into mass spectrometers, and molecular masses of the substrate $[M + 2H] + 2$ ion with $m/z$ 1486.5 and the C-terminal fragment VAYHQLFQEYDNHIKS $[M + H] +$ ion with $m/z$ 1675.5 were detected using positive mode ionization, a capillary voltage of 4000 V, a nozzle voltage of 1500 V, an MS2 scan of 830–1490 $m/z$ over a 330 ms scan time, a fragmentor voltage of 300 V, and a cell accelerator voltage of 3 V. Substrate and product peak areas were determined by integrating the extracted ion chromatograms.

### Mouse NEMO cleavage assay
NEMO peptides (0.2 mg/mL stocks) and 3CLpro (0.5 μM stocks) are in the same assay buffer as described for the human NEMO cleavage assay. Assays were initiated by adding 5 μL enzyme (or buffer for negative controls) to 5 μL peptides. Reactions were incubated at 37 °C in a thermocycler for 30 min and quenched by adding 10 μL quench buffer (50% v/v NuPAGE™ 4× LDS buffer, 20% v/v 0.5 M dithiothreitol, 30% v/v water) and heating at 37 °C in a thermocycler for 20 min. Final reaction concentrations of enzyme and substrate were 0.25 μM and 0.1 mg/mL, respectively. A NuPAGE™ 4–12%, Bis-Tris, 1.0 mm, Mini Protein Gel, 12-well, was loaded with 10 μL/lane and 8 μL SeeBluePlus2 ladder, and proteins were separated with 200 V electrophoresis for 35 min with MES buffer. Bands were visualized with BullDog Bio Acquastain.

### 3CLpro WT expression, purification, and crystallization
BL21(DE3) cells were transformed with pMCSG53 pDNA containing a 3CLpro WT insert with an autoprocessing-sensitive N-terminal Maltose Binding Protein tag and a PreScission protease-sensitive C-terminal His$_6$ tag (provided by Andrzej Joachimiak). Transformants were selected using ampicillin-enriched LB media, grown to OD600 of 0.6–0.8, induced over 10 h with 0.5 mM IPTG (GoldBio, USA), and harvested. Cell pellets were resuspended in 50 mM HEPES pH 7.2, 150 mM NaCl, 5% glycerol, 20 mM Imidazole, 10 mM 2-mercaptoethanol, and lysed by sonication. Insoluble material was pelleted by centrifugation and the lysate loaded onto a Ni$^{2+}$ column, equilibrated with 50 mM HEPES pH 7.2, 150 mM NaCl, 5% glycerol, 10 mM Imidazole, 10 mM 2-mercaptoethanol. The column was washed using 50 mM HEPES pH 7.2, 150 mM NaCl, 5% glycerol, 50 mM Imidazole, 10 mM 2-mercaptoethanol, and eluted using 50 mM HEPES pH 7.2, 150 mM NaCl, 5% glycerol, 500 mM Imidazole, 10 mM 2-mercaptoethanol. The C-terminal His$_6$ tag was cleaved using 1 mg of PreScission to 500 mg of 3CLpro WT at 4 °C, before being flowed through a second Ni$^{2+}$ column, and buffer-exchanged into crystallization buffer 1 (20 mM Tris-HCL pH 8.0, 150 mM NaCl, 1 mM TCEP).

Platelike crystal clusters were produced by adding 10 μL of 3CLpro WT at 5 mg/mL to 10 μL of crystallization matrix (30% PEG 3350, 0.1 M Bis-tris propane pH 7.0) well solution in a hanging-drop vapor diffusion setup at 18 °C. Microseeds were generated from clusters using Seed-Beads (Hampton Research, USA). Single crystals were produced by setting up hanging drop crystallization wells at 18 °C with 10 μL of 3CLpro WT and 10 μL of 1:500 dilution microseeds, using 20% PEG 3350, 0.1 M Bis-tris propane pH 7.0 and 4 mg/mL 3CLpro WT protein concentration, and 20% PEG 3350/0.1 M Bis-tris pH 6.5, 5 mg/mL 3CLpro WT protein concentration.

## 3CLpro C145S expression, purification, and crystallization
BL21(DE3) cells were transformed with pMCSG53 pDNA containing a 3CLpro insert (provided by Andrzej Joachimiak) with a TEV-sensitive N-terminal His$_6$ tag. Transformation, expression and harvesting was performed as with WT 3CLpro. Initial Ni$^{2+}$ column purification was performed as with WT 3CLpro. Elution fractions with highest 3CLpro C145S concentration were combined and dialyzed overnight against 2 L of dialysis buffer (50 mM HEPES pH 7.2, 25 mM NaCl, 5% glycerol, 10 mM 2-mercaptoethanol). 48-hour TEV digestion was initiated by adding 25 μg of TEV per μg of 3CLpro C145S. The solution after cleavage reaction was passed through a second, 6 ml, pre-equilibrated nickel column. The flowthrough from this column was collected and buffer-exchanged using 10,000 MWCO centrifugal concentrators (EMD Millipore, USA) into crystallization buffer 2 (20 mM HEPES pH 7.2, 150 mM NaCl, 2 mM DTT).

3CLpro C145S was incubated overnight with a 0.01-fold molar ratio of Human NEMO capped peptide and incubated overnight at 4 °C. No precipitation was observed. TOP96 (Rigaku Reagents, Japan), BCS (Molecular Dimensions, UK) and GRAS2 (Hampton Research, USA) screens were run using a Gryphon (Art Robbins Instruments, USA). 0.2 μL of protein solution at 6.5 mg/mL was added to 0.2 μL of crystallization matrix in a sitting-drop vapor diffusion setup at 18 °C. BCS screen condition A9 (0.1 M MES pH 6.5, 20% v/v PEG Smear High) produced needle-like crystal clusters of 3CLpro C145S.

## 3CLpro C145S-NEMO co-crystallization
Human NEMO capped peptide (residues 226-235, acetyl-KLAQLQ-VAYH-amide, synthesized by Thermo Scientific) was dissolved to 2 mM in peptide buffer (20 mM HEPES, 150 mM NaCl, pH 7.2) and DMSO (5.67%). Peptides were then added to 3CLpro C145S at a 20-fold molar excess, before overnight incubation at 4 °C. The mixture was centrifuged to remove precipitants, and the supernatant was concentrated to 6.5 mg/mL for crystallization screens.

TOP96 (Rigaku Reagents, Japan), BCS (Molecular Dimensions, UK) and GRAS2 (Hampton Research, USA) screens were run using a Gryphon (Art Robbins Instruments, USA). 0.2 μL of protein solution at 6.5 mg/mL was added to 0.2 μL of crystallization matrix in a sitting-drop vapor diffusion setup at 18C. GRAS2 screen condition F11 (0.1 M Sodium phosphate dibasic dihydrate, pH 9.3, 10 mM Calcium chloride, 20% w/v Polyethylene glycol 3,350), yielded platelike single crystals. These crystals were isolated, cryoprotected by supplementation with 20% glycerol, mounted into ALS style, 0.05-0.1 mm loops (Hampton Research, USA) and flash frozen in liquid nitrogen.

## Data collection and structural determination and refinement
The diffraction data for 3CLpro WT and C145S with and without NEMO were collected at 100 °K at BL12-2 of the Stanford Synchrotron Light Source, using Pilatus 6M detectors and Blu-Ice software[56]. Crystals of 3CLpro WT and NEMO-bound 3CLpro C145S were cryo cooled using their specific mother liquor well solutions supplemented with 30% PEG 3350. Crystals of the C145S were cryo cooled using its mother liquor well solution supplemented with 20% glycerol. WT 3CLpro, NEMO-bound 3CLpro C145S, and 3CLpro C145S data sets were each collected from multiple parts of the crystals. 3CLpro WT and C145S data sets

were successfully merged and processed with XDS[57]. 3CLpro WT was phased with MOLREP[58], using the coordinates of 3CLpro bound to Telaprevir (PDB ID 7K6D)[59]. 3CLpro C145S was also phased using MOLREP, using our 3CLpro WT coordinates. Iterative rounds of manual fitting using COOT[60] and refinements with REFMAC[61] were performed for both structures. Final rounds of refinement for 3CLpro WT and C145S structures resulted in the 1.45 Å and 2.47 Å resolution structures respectively. However, initial data analysis of individual single datasets and combined multiple data sets from NEMO-bound 3CLpro C145S crystals did not yield structure solutions of sufficient quality. Hence, BLEND software[62] was used to merge the best two datasets to 2.50 Å resolution. Next, the structure was solved by molecular replacement with MOLREP, using our 3CLpro WT coordinates. The electron density map showed clear density for the NEMO peptide in two monomers. The active sites for the other two monomers were occupied by the C-terminal tails of neighboring monomers. The NEMO peptides and the C-terminal tails were manually built into the electron density. Several cycles of manual building and refinement were carried out to incorporate other changes in the structure. We next located the water molecules and refined the structure using Phenix 1.20.1[63]. The details of data collection and refinement are given in Supplementary Table 7.

## Preparation NEMO-bound models of 3CLpro from different *betacoronaviruses*
Chains A and B from the crystallographic structure described here were used as the starting configuration for the simulations of the NEMO-bound dimeric SARS-CoV-2 3CLpro. Along with that, structures of 3CLpro from SARS-CoV (PDB_ID 5B6O), MERS-CoV (PDB_ID 5C3N), and HKU1-CoV (PDB_ID 3D23) were used[40,64,65]. After structural alignment, the coordinates from NEMO (aa. 227-234) from the X-ray structure of the NEMO-bound SARS-CoV-2 3CLpro C145S variant were used to build the models of NEMO-bound SARS-CoV, MERS-CoV, and HKU1-CoV 3CLpro dimers. In SARS-CoV-2 3CLpro, Val232 was replaced by Ala232 to also study the interactions in the binding site of 3CLpro with mouse NEMO$_{227-234}$. For the simulations of ligand-free SARS-CoV 3CLpro, PDB_ID 2DUC[40] was used as the starting structure and simulations of ligand-free SARS-CoV-2 3CLpro that were performed prior to generating our crystal structures used PDB_ID 7JUN[13] as the starting structure. NEMO N- and C-termini were capped with acetyl and N-methylamide, respectively. More details about system preparation are described in the Supplementary Methods.

## Molecular dynamics simulations and conformation selection for binding affinity predictions
Different protocols of simulation were conducted for 3CLpro-NEMO$_{227-234}$ and 3CLpro-NEMO$_{190-270}$ systems, as the former was used both for traditional MD analysis and to select conformations for binding affinity predictions (Supplementary Methods). For the 3CLpro-NEMO$_{227-234}$ systems, which were mostly generated in silico, we applied a strategy of MD simulations that aims structural refinement and sampling conformations that visit key interactions. Snapshots selected from these simulations were used as input for machine learning-based binding affinity prediction (MD/ML approach). Similar to the method of protein structure refinement described in Heo et al.[66], sampling was accelerated in a controlled manner by applying a fairly high temperature and weak position restraint potentials that compensate for the high thermal energy, which could drastically propagate the effects of any local molecular distortions or bad contacts. The applied position restraints are much weaker than the energy range of typical non-covalent interactions and conformational changes so that sampling is minimally biased while unrealistic states are easily escaped.

For the MD/ML approach, five independent conformational sampling runs were performed for 224 ns, recording coordinates every 40 ps. Supplementary Methods and Supplementary Table 8

summarizes this MD-based protocol of sampling for conformation selection. The conformations generated from the restrained MD simulations were grouped via a root-mean-square deviation-based algorithm[67]. Considering only $C_\alpha$ atoms of NEMO and residues in the Domains I and II of the of 3CLpro (aa. 16-198), the RMSD of atom-pair distances was computed applying the 1.0 Å cutoff as parameter for clustering. The interaction energy between 3CLpro and the NEMO-peptide was computed using the energy plugin within GROMACS[68]. For each system, three structures with the lowest interaction energy were selected from different conformation clusters within the 20 most populated and used as input for ML-based binding affinity prediction.

In parallel, for classical molecular dynamics analysis, the equilibration phase was fully performed at 310.15 K and five independent production unrestrained runs of 112 ns were carried out. In particular, to check if longer simulations would capture a complete detachment of the NEMO peptide and compare binding stability, the unrestrained MD simulations of SARS-CoV-2 3CLpro bound to mouse and human $NEMO_{227-234}$ were extended to 192 ns. Please see the Supplementary Methods for more details about the simulation parameters and Supplementary Tables 8, 9, which summarize the steps used for classical MD simulations.

### Binding affinity from quantum mechanics calculations

The inclusion of QM or ML in high-throughput drug screening to narrow down the list of promising inhibitor candidates has emerged as a very promising protocol. Recently, we illustrated an encouraging application of QM to refine the binding affinity of classical docking results of COVID-19 spike protein inhibitor drugs[69]. In this work, we employed a similar strategy to evaluate the binding affinity of NEMO with different 3CLPro targets. For the QM-based prediction, we employed the linear-scaling FMO-DFTB[31,32] method to predict binding affinities of NEMO with the four 3CLPro targets. In addition, the polarizable continuum model (PCM) was used to include solvent effects. The PCM was used to describe interactions with solvent, and the empirical D3 dispersion correction was used to improve the accuracy in describing non-covalent interactions. For the PCM calculations, the cavity was calculated using newly calibrated atomic radii to improve accuracy in predicting solvation free energy of DFTB in an aqueous solvent. The details of the atomic radii optimization will be published elsewhere. For each target, the FMO-DFTB/PCM method was used to re-optimize structures of NEMO model systems in bound and unbound states, while the target's structure was fixed. Partial re-optimization of the target did not significantly alter the result. The internal binding energy is defined as the difference in internal energy between the complex and its corresponding unbound NEMO and unbound 3CLPro. We note that QM-calculated internal binding energies can be significantly larger than experimental binding free energies due to the lack of entropy contributions (Supplementary Table 3). Thus, the QM-based binding energies should not be considered an absolute measure of binding affinity but instead a scoring function to rank relative binding affinities[70].

### Binding affinity from machine learning-based rescoring

Machine-learned models dedicated to predicting binding affinities (Demerdash, 2022, in press) were trained on a large database of protein-small molecule complexes with both experimental binding affinities and X-ray crystal complexes known as PDBbind[71], followed by testing on an independent data set, the CASF-2013 benchmark[71], that was not used in the training. For each complex a set of seven features were calculated, where each feature is itself a predicted affinity or free energy calculated under very different physical assumptions (Supplementary Methods). Models were trained using either support-vector machines, gradient-boosted trees, or random

forest. For each of these three methods, models were trained using normalized, or scaled, features. For the gradient-boosted tree and random forest method, models were additionally trained on the raw, unscaled features. For each *betacoronavirus* 3CLpro, the models were applied to the energy minimized complexes and to the set of MD-generated conformers. Separately, the full set of MD conformers with the lowest molecular mechanics (MM) energy of interaction were considered for each system. The average predicted affinity was obtained considering the full set of MD conformers (Fig. 4e and Supplementary Table 3). Noticeably, the ML models applied to energy minimized complexes (conformers prior to MD simulations) did not provide consistent results among the rankings (Supplementary Table 4). These structures have appreciably higher MM energy than the MD structures, suggesting that the consistent rankings of the MD structures reflect more native-like structures, and that the MD protocol of confirmation selection is effective to account for the conformational flexibility of the receptor. The values of the predicted affinities (Supplementary Table 3) are unitless, as they are expressed as $-log(K_d)$. It should be noted that, similarly to the QM predictions, the ML-predicted affinities are used for rankings and the absolute affinities may not be directly comparable with experimental values. The reason for this is that the training database consists largely of proteins bound to relatively small, drug-like ligands. The NEMO peptide is, in general, much larger than the ligands found in PDBbind. Therefore, the predicted affinities of NEMO binding are incommensurate with the binding affinities of the training data. Nonetheless, the physicochemical space covered by PDBbind is vast and is representative of that found at the NEMO-3CLpro binding interface.

### Molecular re-docking of 3CLpro C-terminus

Chains A and B from the crystallographic structure of SARS-CoV-2 3CLpro-NEMO described in this study were used to estimate the contributions of local interactions to the binding affinity of the C-terminus to the dimer interfacial site. The C-terminus of chain B sits between Domains I in the dimer formed with chain A. Chains A and D were used as receptor and ligand, respectively, to estimate the contributions of local interactions to the binding affinity of the C-terminus to a catalytic site in a neighboring dimer. For the calculations of C-terminus at the interfacial and catalytic binding sites, a peptide comprising residues 302–306 was defined from the C-terminus of chain B and D, respectively, detaching it from the rest of the protein by deleting residues 300 and 301. N- and C-termini of these peptides were capped with acetyl and N-methylamide, respectively. Molecular re-docking of the C-terminus peptides was performed using AutoDock Vina[72]. All bonds of the peptide were kept rigid as the goal was to preserve the initial conformation and compute the binding free energy using the Auto-Dock Vina scoring function.

### Reporting summary

Further information on research design is available in the Nature Research Reporting Summary linked to this article.

## Data availability

The data that support this study are available from the corresponding author upon request. Structural data that support the findings of this study have been deposited in RCSB PDB with PDB accession codes: 7T2T, 7T2U, 7T2V. Input and output files of the AutoDock Vina calculations, the energy minimized structures used as input in QM and ML calculations, the DFTB-geometry optimized output structures, the molecular dynamics snapshots selected using the MD/ML protocol, the inputs to MD simulations, the table of features used in the ML calculations, and the unaveraged binding affinity predictions are provided in Supplementary Data 1. Source data are provided with this paper.

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

## Acknowledgements

We would like to acknowledge funding from DOE Office of Science through the National Virtual Biotechnology Laboratory, a consortium of DOE national laboratories focused on response to COVID-19, with funding provided by the Coronavirus CARES Act. This work was partially funded from the Laboratory Directed Research and Development Program of Oak Ridge National Laboratory, managed by UT-Battelle, LCC for the US Department of Energy, LOIS:10074, which supported the conceptual work on the NEMO cleavage in animal models for COVID-19 pathology. Funding for human pathogenesis conceptualization was provided by the National Institutes of Health grant 3RF1AG053303-01S2. Use of the Stanford Synchrotron Radiation Lightsource, SLAC National Accelerator Laboratory, was supported by the U.S. Department of Energy, Office of Science, Office of Basic Energy Sciences under Contract No. DE-AC02-76SF00515. The SSRL Structural Molecular Biology Program is supported by the DOE Office of Biological and Environmental Research, and by the National Institutes of Health, National Institute of General Medical Sciences (P30GM133894). The contents of this publication are solely the responsibility of the authors and do not necessarily represent the official views of NIGMS or NIH. We also thank Daniel Fernandez and the Macromolecular Structure Knowledge Center at Stanford for providing equipment for protein crystallization. This research used resources of the Oak Ridge Leadership Computing Facility, which is a DOE Office of Science User Facility supported under Contract DE-AC05-00OR22725. This research also used resources at the Spallation Neutron Source and the High Flux Isotope Reactor, which are DOE Office of Science User Facilities operated by the Oak Ridge National Laboratory. The Office of Biological and Environmental Research supported research at ORNL's Center for Structural Molecular Biology, a DOE Office of Science User Facility. This manuscript has been authored by UT-Battelle, LLC under Contract No. DE-AC05-00OR22725 with the U.S. Department of Energy. The United States Government retains and the publisher, by accepting the article for publication, acknowledges that the United States Government retains a non-exclusive, paid-up, irrevocable, world-wide license to publish or reproduce the published form of this manuscript, or allow others to do so, for United States Government purposes. The Department of Energy will provide public access to these results of federally sponsored research in accordance with the DOE Public Access Plan (http://energy.gov/downloads/doe-public-access-plan).

## Author contributions

M.A.H. and I.I.M. conducted the crystallography experiments; E.T.P., B.K.A., and M.B. conducted molecular dynamics simulations and molecular docking; O.D. performed machine learning predictions; S.I. and V.-Q.V. performed quantum mechanics calculations; M.I. and S.R.

conducted protein expression experiments; D.W.K. and A.K. assisted with crystallographic analysis; J.C.M. and E.T.P. performed protein hot spot predictions; A.L. and S.G. conducted the cleavage assays and mass spectrometry experiments; M.G. conceived the initial hypothesis; M.A.H., E.T.P., I.I.M, O.D., V.-Q.V, S.R., S.G., and M.G. wrote the manuscript. S.W. and D.J. initiated the project and guided the entire work.

## Competing interests

The authors declare no competing interests.
