## [Peer Review File · Nature Communications]

Structural and functional characterization of NEMO cleavage
by SARS-CoV-2 3CLproReviewers' Comments:

Reviewer #1:

Remarks to the Author:

This manuscript provides compelling evidence that the host signaling protein NF- κ B essential modulator can be cleaved by SARS-CoV-2 3C-like protease, potentially mediating dysregulation of the host immune response. The authors carry out a combined experimental and computational approach to provide deep mechanistic insights into how the viral protease is able to bind to both host cell and viral proteins with high specificity. For the most part, the manuscript is well-written, methodically carried out, and is of high importance given the ongoing SARS-CoV-2 pandemic. However, there are some areas of the manuscript where the data does not appear to fully support the stated conclusions, and some areas that lack clarity. In general, too much is delegated to the SI, presenting difficulties for following along with the detailed biochemical arguments presented in the main text. Yet, with appropriate revision, this manuscript should be publishable and of broad interest to the readership of Nature Communications.

Comments/questions for clarification:

1. Line 38: define 3CLpro after "protease".
2. Line 57: it is not clear why multiple therapeutic approaches will need to be implemented (simultaneously?) to 'eradicate SARS-CoV-2'. For example, it has been suggested that a universal vaccine targeting conserved regions of viral surface proteins could be a potential solution.
3. Line 58: suggested to write "prevent future zoonotic outbreaks."
4. Line 65: cites reference [23]. Shouldn't this be [2]?
5. Lines 59-60: please include references to support that this is an 'effective means...'
6. Lines 65-66: the authors could consider citing the success of protease-targeting therapeutics against other highly mutable pathogens such as HIV to bolster their claim/argument.
7. Lines 69-79: it is difficult to follow along here without some mechanistic schematic/figure; might one be included?
8. Line 91: remove "from".
9. Line 98: it is stated that this is the "first structure of SARS-CoV-2 3CLpro that is bound to a human peptide substrate". Are there existing structures of SARS-CoV-2 3CLpro bound to non-human peptide substrates?
10. Lines 109-112: these lines posit that much of the current work has already been done. A few sentences here distinguishing the novelty of the current work would be helpful/informative.
11. Line 129: should say "analysis of human NEMO (hNEMO)", I believe?
12. Line 129-130: five amino acids are listed as potential 3CLpro recognition sites. Why did the authors narrow in on Gln231?
13. Lines 132-133: please define GST, and are these duplicates of the previous line (i.e., GST-96-250 vs. 96-250, etc.). Also, the previous lines states five constructs were explored, but it looks like only three ranges of amino acids are listed (96-250, 221-250, 221-339)?

14. Figure 1: Please define UBAN. Also, I cannot find where Fig. 1c is referenced in the text. Also, since there are 16 stated nsps in SARS-CoV-2, what does Nsp10/22 refer to?
15. Line 208: it is confusing to refer to different regions of a single substrate as a separate substrate, as in "substrate P2".
16. Lines 200-201: I find the title of section 2.4 to be confusing, as this section really just discusses the positioning of glutamine at P1, rather than substrate versatility. It seems that this case cannot be made until the next section, in which binding of NEMO is compared to that of the C-terminus. Perhaps the subsection titles can be adjusted to better reflect the text?
17. Lines 236-237: I am not convinced that it can be definitively stated that since 'S3 and S1 subsites share the same interactions, all the listed interactions are required for substrate-binding in 3CLpro'. Perhaps this is just coincidental; it would seem that testing the binding of other similar substrates with slightly altered interactions would be necessary to conclude this.
18. Line 242: it is not clear what "such interactions" refers to, and what data there is to support this statement.
19. Sections 2.6 and 2.7: difficult to follow since the reader needs to flip back and forth between the main text and SI so frequently. Perhaps move Fig. S3 to the main text? Also, in general, I find section 2.7 to be quite confusing. It is difficult to distinguish if the noted differences in packing are due to differences in c-term binding location (interfacial vs. non-interfacial grooves) or to the presence of bound NEMO, or both.
20. Lines 271-278: why was docking used to calculate binding affinities, as opposed to MD? Binding affinities from docking – especially rigid body docking – are usually not very accurate. Are the results statistically significant (implied by "by far" – line 274)? The authors might consider running MD on the interfacial docked conformation and computing binding affinities via MD (e.g., mmpbsa). It is also not clear why the binding affinity results lead to the stated conclusion of decreased self-inhibition; this argument should be strengthened.
21. Section 2.8: though included later in the methods, a few sentences here describing the simulation methods and what simulations were done would be helpful for the reader (e.g., it looks like five trials were run of each simulation type from Fig. 3? How was the complex with the longer construct of hNEMO generated? Docking? Crystal structure?).
22. Lines 327-332: what "competing interactions" are new with V232 that were not there with A232? This argument is not clear to me.
23. Lines 345-346: are these small differences in RMS really significant?
24. Lines 358-359: it is stated that possibly, "3CLpro affects the conformational flexibility of the binding site, which may be captured in MD/ML but not in the QM approach". Could this be assessed with MD?
25. Line 390: the first two sentences sound odd/incomplete next to one another.
26. Lines 441-444: In line with one of my previous comments, I am not convinced the data supports the statement that the simulations show "increased local rigidity" due to the S46A mutation (and shouldn't the A lead to higher rigidity than the S, according to lines 445-448?). Also, note that SARS-CoV-2 is listed twice on lines 442-443 (one should be SARS-CoV maybe?).

Reviewer #2:

Remarks to the Author:

Hameedi et al. provide structural evidence for the interaction of an inactive mutant of SARS-CoV-2 3CLpro and the human NF- κ B Essential Modulator (NEMO) in combination with extensive in silico analysis they suggest that the cleavage of NEMO by SARS-CoV-2 3CLpro explains the virus fitness in humans. The novelty of these findings is partly limited, as the processing of NEMO by 3CLpro and its physiological relevance for SARS-CoV-2 infections in the brain has recently been reported by Lampe et al (ref. 21) and the closely related structure of porcine epidemic diarrhea virus 3CLpro in complex with NEMO has been described by Ye et al. (ref. 16) as well.

The manuscript is generally well written, however, in parts it could be more concise. Further the nomenclature should be made consistent throughout the manuscript, in particular for NEMO peptide fragments. Most figures showing protein structures are too crowded to clearly convey their message and should be simplified. For example, the surface representations should be displayed in one color only.

A concern is the very moderate peptide cleavage efficiency of NEMO by 3CLpro as reported in Figure 1, reaching only about 12% after 2 h incubation at a NEMO concentration of 13 μ M and 3CLpro concentration of 5 μ M. The authors should at least discuss these numbers and their relevance in physiological conditions. Furthermore the reported structure of 3CLpro in complex with NEMO exhibits an unusually large Rfree gap indicating potential model bias. The authors should provide omit maps for the bound NEMO peptides to support their binding hypotheses.

Some more detailed comments:

l 65: citation should read "2,3"

l 91: "form" instead of "from"?

l 137: Maybe reference 21 (Lampe et al. The SARS-CoV-2 main protease Mpro causes microvascular brain pathology by cleaving NEMO in brain endothelial cells. *Nat. Neurosci.* 24, 1522-1533.) should already be cited in the introduction to underline the physiological importance of NEMO cleavage by 3CLpro.

l 139: Figure 1, overall low cleavage efficiency. Was a longer incubation attempted and, was 100% cleavage reached? If not, what was the reason? Can the authors exemplarily show the analysis of the raw data (HPLC chromatogram/MS analysis)?

l 146: Figure 2, the caption does not clearly explain the figure. For example, the NEMO peptide is shown with a mesh. However it is not clearly written what kind of electron density is depicted (2Fo-Fc?), also the contour level as well as the carving radius must be stated.

l 167: Please specify the NEMO fragment in a consistent way.

l 169: "Chain B and C are each bound [...] (Figure 2a)." Chain B is not depicted in Figure 2a.

l 192-198: To which wt structure do the authors compare their complex structure? If this is the wt structure also reported in the manuscript, this should be mentioned. An additional figure to demonstrate the conformational changes would be helpful here. It might also be useful to take advantage of the many wt structures of 3CLpro that are already in the PDB to see if these conformational changes are indeed induced through the peptide or are also observed in other structures.

l 209-210: It would be helpful to show all the mentioned pockets in Figure 2. For this a unicolored surface would be more insightful than the current illustration.

l 228-242: please compare also with other 3CLpro/peptide complexes, for example PDB 7KPH and 7JOY from Lee et al. (ref. 10 in manuscript) and 7N6N (N- and C-terminus bound to 3CLpro) from Noske et al (ref. 29). Are the interactions observed in the author's structure also seen in these structures? It would also be helpful to extend Table 1 by the interaction of the 3CLpro C-terminal peptide with the enzyme to allow for better comparison with the NEMO-specific interactions. Furthermore the NEMO/3CLpro complex and its binding mode should also be compared to the structure of SARS-CoV 3CLpro with a longer N-terminal substrate (PDB 2Q6G) to derive a more general binding model for peptides to 3CLpro.

I 246/Suppl. Fig S3: S3a, in inset the figure would become better interpretable when the C-terminus is not shown as surface but only as stick model.

I 246-253: This section needs to be supported by an additional supplementary figure showing the proposed binding pocket and interactions. This becomes not clear from Fig S3a.

I 258: Please specify the NEMO fragment in a consistent way.

I 259 ff: "Arg298 of chain B is closer...". A supplementary figure would be helpful to illustrate this comparison.

I 274-276: "These comparable binding affinities...". Is there precedent for this type of double binding site for decreasing self-inhibition? Or is this unique property of SARS-CoV-2 3CLpro? The authors should state that these affinities are predicted or otherwise try to experimentally validate their statement.

I 291, figure 3: Panel c is missing labels for the x-axis. Also please indicate the active site on 3CLpro to guide orientation of the reader. Furthermore, please explain the discrepancy between the run productive run length mentioned of 112 ns in the Materials and Methods section and the graph at a run length of 190 ns. Also, please specify the NEMO fragment in a consistent way.

I 361: Visualization of results in Fig. 4b should be improved. Can the results of the MD/ML affinity prediction be quantified in a similar manner to the QM methods? Otherwise give the results for the individual methods in the supplement.

I 379: In addition to the binding of NEMO to 3CLpro mostly governed by their calculated affinity, it is worthwhile to also include a more detailed analysis of the NEMO coiled coil structure of the template. For example, TWISTER analysis (Strelkov & Burkhardt 2002, PMID 12064933) of PDB model 6MI3 suggests a coiled-coil anomaly just prior to this unfolded region, which could be advantageous for CC dissociation. This increases the interhelical distances and thus destabilizes the hydrophobic core of the coiled coil. This might be worth discussing to better argue why this might be a preferred site for initial unwinding and subsequent 3CLpro binding and cleavage

I 402: Figure 5, black line seems of lower resolution than rest of image.

I 415: Please state you compare a structure solved at cryo conditions with a structure solved at room-temperature.

I 703-705: Could the authors clarify why different run length parameters were used for the different 3CLpro and NEMO complexes.

Supplement:

I 89: Please explain the different coloring of the table text in the table legend.

I 100: Resolution limits of outer shells are missing. Please also state the overall I/sigma and not only I/sigma for the outer shell.

I 112: Figure S1, please also state the identity between mouse and human NEMO to highlight their identity.

I 117: Figure S2 would be more intuitive if the cleavage products of the different NEMO constructs were directly labeled.

I 122: Figure S3a, removal of the surface representation of the C-terminal peptide would make the figure clearer. Also, state the type of electron-density depicted in panel b and c as well as their contour level.

I 134: Figure S4, a scheme as additional panel showing the position of analyzed interchain distances would enhance interpretability of MD results.

I 147: Figure S6, please also show same analysis for SARS-CoV 3CLpro to highlight differences mentioned in the text.

Reviewer #3:

Remarks to the Author:

The manuscript characterizes the cleavage of fragments of the NF- κ B Essential Modulator (NEMO) by the 3CLpro protease of SARS-CoV-2. The authors validate the cleavage at one of the previously reported cleavage sites, Gln231, and solve the structure of a short NEMO peptide (226-235) bound to

the active site mutant C145S of 3CLpro. The complex structure confirms binding pocket characteristics that modulate substrate selectivity. The authors also propose a role for the C-terminal tails of 3CLpro in quaternary structure interactions and self-inhibition. Machine learning- and physics-based computational methods are applied to compare the interaction of NEMO with SARS-CoV-2 and other coronaviruses and its effect on pathogenesis.

The manuscript is well crafted, the methodology is described in detail and the conclusions are in line with the data presented.

Please find detailed below some of the concerns with the manuscript.

Results: 2.1. The cleavage of hNEMO at different sites by SARS-COV-2 3CLpro was previously reported (manuscript ref. 20), as well as the catalytic efficiency of the cleavage at Gln231, which affect the novelty of these findings. The manuscript could benefit from a comment on the other cleavage sites (Q83, Q205, Q304 and Q313) that were not observed in this study. The sensitivity of SDS-PAGE detection may be an issue for low % cleavage.

Results 2.2-2.7. As the authors mention, the Protein data Bank is replete with structures of SARS-CoV-2, including WT apo protein, complexes with peptides, peptidomimetics and small molecule inhibitors, and C145A/S mutants. The complex of a similar NEMO peptide with 3CLpro for Porcine Epidemic Diarrhea virus was also reported (ref. 14). The complex structure in this manuscript highlights the characteristics of the SARS-CoV-2 3CLpro binding site as it interacts with NEMO. The additional insight may further our understanding of the selectivity/versatility of 3CLpro. The structure also details the binding of the C-terminal tail to the substrate binding site and at dimers interfaces. Similar findings were reported previously. The description of the C-terminus binding at the dimers interface (2.7) could benefit from a figure, it is detailed but difficult to follow. Supplementary Figure S3 does not show all the described interactions and the two chains are depicted in similar shades of pink.

The statistics from the structure refinement indicate $R_w/R_f = 0.214/0.309$. The R_{free} value is high for a 2.14 Å dataset, and the large spread between R_w and R_f may indicate that the structures were over refined. The authors should try an alternative refinement strategy. The same is valid for the 3CLpro C145S structure ($R_w/R_f = 0.194/0.273$).

Results 2.8-2.9. The molecular dynamic simulations examine the interaction of a longer NEMO fragment (190-270) with 3CLpro. The model for the long NEMO fragment was built using the NEMO structure from PDB 6MI3 as a template (the model also appears to be used in the analysis of hydrophobic interactions within the NEMO dimer, discussion). 6MI3 reports the structure of a different domain of NEMO (51-112) fused to long N- and C-terminal ideal coiled-coil adapters. Both portions seem inadequate: the GCN4-derived portion and the NEMO portion, which due to the low coiled-coil propensity in the central sequence adopts a peculiar structure. The crystal structure of NEMO(150-272) was reported (in complex with a viral FLIP) and may be a better starting point for the simulations (3CL3). Also the manuscript would benefit from more detail on how the "partially unfolded" NEMO(190-270) structure was generated and the comparison with dimeric NEMO.

The differences reported in 2.9 lines 325-327, comparing mNEMO and hNEMO, seem to be within the reported errors: the authors should add a comment on why they are relevant or remove.

Simulations utilizing the experimental complex structure of this work and comparative analysis of betacoronaviruses: a comment on the impact that starting from an experimental complex structure (NEMO+ SARS-CoV-2) versus a model (NEMO + other coronaviruses 3CLpro) may have on the predicted binding affinity may be helpful.

Figure 5 is difficult to read due to the coloring of the residues (same N, O and S), and possibly the lack of a substrate.

Methods:

Line 613: "25 ug of TEV per ug of 3CLpro", correct "u" for "μ" or "m"?

Lines 621-627: in two different paragraphs two different descriptions of how the peptide/3CLpro

crystals were prepared.

Also:

Line 91: "The structure of 3CLpro from in complex" is not clear

Lines 392-393: "non covalent associations of the IKK α / β domains of NEMO" is not clear what the authors mean.

Lines 442-443: one is meant to be SARS-CoV rather than SARS-CoV-2?

Supplementary Table S3: superscripts a and b are used in the table but not explained.

Supplementary Figure S2: the expected cleavage band from NEMO(232-250) is not observed in lane 6 (it is observed elsewhere). The band is likely weak because of the small size and the lane may have a lower loading than the others?

REVIEWER COMMENTS / Authors' responses in blue font

Reviewer #1 (Remarks to the Author):

This manuscript provides compelling evidence that the host signaling protein NF- κ B essential modulator can be cleaved by SARS-CoV-2 3C-like protease, potentially mediating dysregulation of the host immune response. The authors carry out a combined experimental and computational approach to provide deep mechanistic insights into how the viral protease is able to bind to both host cell and viral proteins with high specificity. For the most part, the manuscript is well-written, methodically carried out, and is of high importance given the ongoing SARS-CoV-2 pandemic. However, there are some areas of the manuscript where the data does not appear to fully support the stated conclusions, and some areas that lack clarity. In general, too much is delegated to the SI, presenting difficulties for following along with the detailed biochemical arguments presented in the main text. Yet, with appropriate revision, this manuscript should be publishable and of broad interest to the readership of Nature Communications.

We thank you for the thorough review of our manuscript and for the important points raised, which enabled us to improve clarity of this study. As detailed below, we have transferred selected figures from the SI to the main text to facilitate following along our discussion. Our responses to the Reviewer's comments and questions are written in blue font. In the manuscript, all the changes are tracked and the lines of those related to the reviewer's comments are informed in each response. Please note that, following the Nature Communications formatting instructions, the subheadings in Results are no longer numbered and the subheadings in Discussion were removed.

Comments/questions for clarification:

1. Line 38: define 3CLpro after "protease".

We have updated the text with the definition of 3CLpro (line 74) accordingly.

2. Line 57: it is not clear why multiple therapeutic approaches will need to be implemented (simultaneously?) to 'eradicate SARS-CoV-2'. For example, it has been suggested that a universal vaccine targeting conserved regions of viral surface proteins could be a potential solution.

Thank you for pointing this out for clarification. We have updated the text (lines 91-111) to clarify that the radically uneven pace of vaccination is an important obstacle to reach herd immunity and that the progressive emergence of variants with enhanced fitness has proven that SARS-CoV-2 eradication is extremely challenging (Kofman et al. 2021). Therefore, the circulation of SARS-CoV-2 in the population is likely to persist at various levels in different geographical regions and it is imperative to establish more broadly protective vaccines and effective therapeutic approaches to steadily reduce the risk of severe illness.

3. Line 58: suggested to write "prevent future zoonotic outbreaks." We have updated the text as suggested (lines 111).

4. Line 65: cites reference [23]. Shouldn't this be [2]? Thank you for identifying this inconsistency in the document. It should be [2,3]. We have corrected it (line 113).

5. Lines 59-60: please include references to support that this is an 'effective means...'. We have added references that describe antiviral strategies targeting the disruption of the virus' life cycle. We have modified the sentence to not emphasize the effectiveness of these treatments (lines 112-113), since this would involve deepening a discussion (e.g., viral adaptation to promote drug resistance) that is beyond the scope of our study (Neagu et al., 2018, DOI: 10.1371/journal.pcbi.1005947).

6. Lines 65-66: the authors could consider citing the success of protease-targeting therapeutics against other highly mutable pathogens such as HIV to bolster their claim/argument. We have now cited the PaxlovidTM, from Pfizer, which is a drug cocktail that includes the component nirmatrelvir, which targets the SARS-CoV-2 main protease and had the emergency use recently approved by the FDA (lines 118-121).

7. Lines 69-79: it is difficult to follow along here without some mechanistic schematic/figure; might one be included? We agree it is difficult to follow without a schematic representation of the reaction mechanism. However, as the details of the reaction mechanism is not the focus of our study, we decided to simplify this paragraph and cite Ramos-Guzmán et al., 2020, which presents a thorough analysis the reaction mechanism of proteolysis catalyzed by SARS-CoV-2 3CLpro (<https://doi.org/10.1021/acscatal.0c03420>) (lines 126-131).

8. Line 91: remove "from". OK (line 186)

9. Line 98: it is stated that this is the "first structure of SARS-CoV-2 3CLpro that is bound to a human peptide substrate". Are there existing structures of SARS-CoV-2 3CLpro bound to non-human peptide substrates?

Thank you for the comment and the question. There are structures of viral peptide substrates bound to SARS-CoV-2 3CLpro. In fact, one (3CLpro bound to Ac-SAVLQSGF-CONH₂, PDB ID: 7N89) is published by our co-author, D.W. Kneller. We have added references to several structures that are bound to either their N- or C-terminal viral autoprocessing sequences, including 7N89, 7JOY and 7N6N, throughout the text for the purpose of comparison to our NEMO-bound structure (lines 1293-1428 and Supplementary Tables S1 and S5).

10. Lines 109-112: these lines posit that much of the current work has already been done. A few sentences here distinguishing the novelty of the current work would be helpful/informative. We have moved those sentences from Results to the Introduction, where we clarify the novelty and relevance of this work (lines 173-175, 227-222). We also added to the Introduction a reference to Wenzel et al (ref. 19 - the first author name was corrected), which demonstrates the biological relevance of NEMO cleavage by 3CLpro and supports the importance of our study in understanding the molecular details underpinning this process. Our study is unique in unraveling key structural features of 3CLpro-NEMO binding and predicting their connections with differences in pathogenesis across species.

11. Line 129: should say “analysis of human NEMO (hNEMO)”, I believe? Yes, thank you. We have also changed the sentence to refer to Wenzel et al., in which the five 3CLpro recognition sites in hNEMO are detected *in vitro* (line 269).

12. Line 129-130: five amino acids are listed as potential 3CLpro recognition sites. Why did the authors narrow in on Gln231? Cleavage at the five recognition sites was detected in Wenzel et al. (<https://doi.org/10.1038/s41593-021-00926-1>). However, we have performed gel assays of 3CLpro incubated with long constructs of mouse NEMO that include four of the potential 3CLpro recognition sites, namely, Gln205, Gln231, Gln304, and Gln313. In this experiment, we only observed substantial cleavage at site Gln231, suggesting that it is more susceptible to proteolysis (Supplemental Figure S2). We have updated the Supplemental Fig. S2 to identify the proteolysis products and added a sentence to clarify this point (lines 279-281). In addition, we further discussed the potential reasons for a preferential cleavage at Gln231 using analyses of the coiled-coil structure of NEMO and predictions of molecular recognition features (MoRFs) on NEMO that undergo disorder-to-order transition upon binding. We hypothesize that the combination between a lower propensity for coiled-coil formation and the presence of MoRFs at this unfolded region of NEMO contributes to the preferential cleavage at site Gln231 by 3CLpro. Please see lines 1063-1073, 1211-1232, and Fig. 3c.

13. Lines 132-133: please define GST, and are these duplicates of the previous line (i.e., GST-96-250 vs. 96-250, etc.). Also, the previous lines states five constructs were explored, but it looks like only three ranges of amino acids are listed (96-250, 221-250, 221-339)? We have adjusted this sentence to clarify these questions. 3CLpro enzyme was incubated with three constructs of NEMO, with and without a glutathione S-transferase (GST) tag, namely, a.a. 96-250, 221-250, 221-339, GST-96-250, GST-221-250, and GST-221-339 (lines 232-233).

14. Figure 1: Please define UBAN. Also, I cannot find where Fig. 1c is referenced in the text. Also, since there are 16 stated nsps in SARS-CoV-2, what does Nsp10/22 refer to? UBAN is now defined in Fig. 1a. Fig. 1c is referenced (line 224 and 619), and NSP10/22 has been changed in Fig. 1c to Nsp10/11.

15. Line 208: it is confusing to refer to different regions of a single substrate as a separate substrate, as in “substrate P2”. We have corrected it (line 450).

16. Lines 200-201: I find the title of section 2.4 to be confusing, as this section really just discusses the positioning of glutamine at P1, rather than substrate versatility. It seems that this case cannot be made until the next section, in which binding of NEMO is compared to that of the C-terminus. Perhaps the subsection titles can be adjusted to better reflect the text? We have changed the title from “Polar and hydrophobic subsites in 3CLpro position a glutamine at P1 while promoting substrate versatility” to “Eight 3CLpro subsites underpin substrate-binding and selectivity” (line 442). We note, however, that we start discussing the basis of the substrate versatility of 3CLpro in this subsection by pointing out the predominance of H-bond interactions with the backbone of hNEMO in the S4-S2 and S1’-S4’ subsites. We also highlight the role of hydrophobic interactions in substrate-binding selectivity of 3CLpro in Table 1 (Lines 422-627).

17. Lines 236-237: I am not convinced that it can be definitively stated that since ‘S3 and S1 subsites share the same interactions, all the listed interactions are required for substrate-binding in 3CLpro’. Perhaps this is just coincidental; it would seem that testing the binding of other similar substrates with slightly altered interactions would be necessary to conclude this. We have changed the sentence to say that only interactions in the S3 subsite are conserved between the two. We removed any mention of requirement for substrate-binding at this subsite. Furthermore, we observe different interactions in S1, S2 and S4 subsites, which therefore may account for the versatility of this enzyme (Fig. 1c, Table 1) (lines 630-632). A broad comparison with other structures was also performed and is consistent with that (Supplementary Figure S5, Lines 1078-1108).

18. Line 242: it is not clear what “such interactions” refers to, and what data there is to support this statement. We have amended the manuscript to reflect that the interactions we are referring to are those in the S1’-S4’ subsites of 3CLpro that are potentially key for substrate versatility of 3CLpro (lines 670-671). We discuss how S and S’ subsites contribute to the substrate versatility of 3CLpro in the Discussion Section of the manuscript). This involves a structural comparison with many published structures to determine how each subsite changes in binding different substrates (Lines 1078-1108, Supplementary Figure S5).

19. Sections 2.6 and 2.7: difficult to follow since the reader needs to flip back and forth between the main text and SI so frequently. Perhaps move Fig. S3 to the main text? Also, in general, I find section 2.7 to be quite confusing. It is difficult to distinguish if the noted differences in packing are due to differences in c-term binding location (interfacial vs. non-interfacial grooves) or to the presence of bound NEMO, or both.

We thank the reviewer for their suggestions and comments. We have consequently moved Fig. S3 to panels b and e of Fig. 2 and improved the diagram representing the interfacial C-terminal tail binding site (Fig. 2e). Regarding Section 2.7 (now titled “Interfacial C-terminus may attenuate self-inhibition in 3CLpro dimer”, line 684), we have decided to remove the analysis of interactions formed by Arg298 as the positions of its guanidinium N atoms are not well defined in our crystal structure; and we believe this deletion does not impact our findings described in the manuscript. In this subsection, we focus on the analysis of the predicted relative binding affinities of the hNEMO peptide in the catalytic site (Fig. 2c) and C-terminus in the interfacial (Fig. 2e) and catalytic binding sites (Fig. 2d). Please see our response to the Reviewer’s next question #20.

20. Lines 271-278: why was docking used to calculate binding affinities, as opposed to MD? Binding affinities from docking – especially rigid body docking – are usually not very accurate. Are the results statistically significant (implied by “by far” – line 274)? The authors might consider running MD on the interfacial docked conformation and computing binding affinities via MD (e.g., mmpbsa). It is also not clear why the binding affinity results lead to the stated conclusion of decreased self-inhibition; this argument should be strengthened.

The reviewer is correct in stating that the scores of docking programs are not intended to be compared to experimental binding free energies and are not intended to be used in binding affinity/free energy prediction. Rather, the scores output by a docking program are intended to be used for the express purpose of ranking docked poses to obtain the putative most native-like structure. In the present case, we have the native ligand conformations from the crystallographic structure, and we use AutoDock Vina scoring function to predict qualitatively the *relative* binding affinity of the C-terminal peptide to the catalytic site versus the interfacial site. For the purpose of ranking ligand conformations, AutoDock Vina performs very well compared to other scoring functions (either standalone scoring functions or scoring functions from docking programs) when evaluated on the PDBbind benchmark (*cf.* Gaillard T, *J Chem Inf Model*, 2018, 58: 1697-1706). We have changed the text to clarify that the AutoDock Vina binding scores are used for qualitative and comparative discussions (lines 688-753).

21. Section 2.8: though included later in the methods, a few sentences here describing the simulation methods and what simulations were done would be helpful for the reader (e.g., it looks like five trials were run of each simulation type from Fig. 3? How was the complex with the longer construct of hNEMO generated? Docking? Crystal structure?).

Motivated by suggestions from the other Reviewers and given the limitation of obtaining a more robust model for the simulations of the long construct of NEMO bound to 3CLpro, we opted to use a different approach to explore the effects of coiled-coil propensity on 3CLpro-NEMO binding. In short, our simulations of NEMO(190-270)-bound WT 3CLpro reproduced the interactions captured experimentally and suggested that the competing interactions within the NEMO dimer and those with 3CLpro can affect binding of mouse and human NEMO to 3CLpro differently. However, the only available structure of NEMO in the region of interest is 3CL3 (aa. 150-272) and there, NEMO interacts with FLIP, it may not either well represent the state preceding its binding to 3CLpro. Instead of these simulations, analysis of coiled-coil parameters using TWISTER (Strelkov and Burkhard 2002), complemented by predictions using PAIRCOIL (Berger et al. 1995) is described. ANCHOR (Dosztányi et al. 2009; Oates et al. 2013) was used to predict potential disordered binding regions in NEMO. We hypothesize that the combination between a lower propensity for coiled-coil formation and the presence of molecular recognition features (MoRFs) at this unfolded region of NEMO contributes to the preferential cleavage at site Gln231 by 3CLpro. In addition, we link the mouse NEMO predicted higher coiled-coil propensity near this compared to human NEMO to the lower stability suggested in the simulations of NEMO peptide-bound 3CLpro. Please see lines 1063-1073, 1211-1232, and Fig. 3c. An illustrative representation of the 3CLpro bound to NEMO dimer is kept for visualization (Fig. 3b).

22. Lines 327-332: what “competing interactions” are new with V232 that were not there with A232? This argument is not clear to me.

We suggest that the interactions within the NEMO dimer compete with the interactions between NEMO and 3CLpro. These competing interactions are present both in the complex with human and mouse NEMO (hNEMO and mNEMO, respectively). However, our simulations indicate that they are effective to perturb the complex with mNEMO but not hNEMO, which appears to be more strongly bound to the catalytic site of 3CLpro. This is consistent with the new coiled-coil

analysis using TWISTER and PAIRCOIL. We changed the manuscript to better clarify this argument (lines 855-863).

23. Lines 345-346: are these small differences in RMS really significant?

Even though SARS-CoV-2 3CLpro is highly conserved relative to SARS-CoV 3CLpro (96% identity), it is reported a slightly reduced proteolytic efficiency, a looser dimer packing (Zhang et al., Science 2020), and a small overlap of substrates (Koudelka et al., Proteomics 2021) of SARS-CoV 3CLpro relative to SARS-CoV-2 3CLpro. In addition, we predict via quantum mechanics and molecular dynamics/machine learning-based methods a higher binding affinity of NEMO to SARS-CoV-2 3CLpro compared to SARS-CoV 3CLpro. We then discussed the likely molecular features underpinning functional differences and that are also predicted to affect NEMO cleavage. We do not posit that a small difference in RMS fluctuations, alone, explains it, but that a combination of observations from our structural analysis, molecular dynamics simulations, and previous experimental data suggests that substitutions at sites 285, at the dimer interface, and 46, near the catalytic site are key. In particular, the effect of the substitution S46A in local RMS fluctuations is consistent with the different helical propensities of these residues and reflected in crystal structures of SARS-CoV-2 and SARS-CoV 3CLpro. The latter adopts a secondary structure of 3_{10} helix in the region (Wang et al., Sci Rep 2016, <https://doi.org/10.1038/srep22677>). Finally, *small differences in molecular rigidity observed in molecular dynamics simulations of intrinsically disordered proteins were demonstrated, indeed, to be correlated with major differences in proteolytic efficiency* (Prates et al., Chem Sci, 2018, DOI: 10.1039/C7SC05016J). We have added text in the Discussion to point this out (lines 1112-1115) and to state that these results reinforce the importance of performing site-directed mutagenesis and enzymatic assays to quantify the contribution of non-conserved amino acids in 3CLpro from different species to the efficiency of NEMO cleavage (lines 1195-1209).

24. Lines 358-359: it is stated that possibly, “3CLpro affects the conformational flexibility of the binding site, which may be captured in MD/ML but not in the QM approach”. Could this be assessed with MD?

The MD/ML approach involves restrained MD simulations to refine complexes toward a *holo* conformation of the protein. This protocol was developed based on the strategy of protein structure refinement described in Heo et al (<https://dx.doi.org/10.1021/acs.jctc.0c01238>). A few conformations are selected from the simulations based on population and energetics criteria. We find that this method performs very well to rank binding affinities when combined with machine learning models that are not strictly dependent on a highly accurate model of protein-ligand complexes as long as conformations sampling key interactions are used as input. A more systematic demonstration of the performance of our MD/ML workflow in benchmarking systems is expected to be published soon. In contrast, we have not thoroughly tested a protocol combining MD simulations with QM calculations and are not confident that it would provide better results. More accurate QM calculations may require native-like structures instead of a set of conformations that sample key interactions separately.

25. Line 390: the first two sentences sound odd/incomplete next to one another. As suggested, we have made changes to improve clarity (lines 987-991).

26. Lines 441-444: In line with one of my previous comments, I am not convinced the data supports the statement that the simulations show “increased local rigidity” due to the S46A mutation (and shouldn't the A lead to higher rigidity than the S, according to lines 445-448?). Also, note that SARS-CoV-2 is listed twice on lines 442-443 (one should be SARS-CoV maybe?).

We hope that our answer to question 23 addresses this. We suggest that Ala46 leads to higher rigidity than Ser46. We edited this part of the text for better clarity and corrected the sentence in which SARS-CoV-2 was listed twice (lines 1109-1118).

Reviewer #2 (Remarks to the Author):

Hameedi et al. provide structural evidence for the interaction of an inactive mutant of SARS-CoV-2 3CL_{pro} like protease (3CL_{pro}) and the human NF- κ B Essential Modulator (NEMO) in combination with extensive in silico analysis they suggest that the cleavage of NEMO by SARS-CoV-2 3CL_{pro} explains the virus fitness in humans. The novelty of these findings is partly limited, as the processing of NEMO by 3CL_{pro} and its physiological relevance for SARS-CoV-2 infections in the brain has recently been reported by Lampe et al (ref. 21) and the closely related structure of porcine epidemic diarrhea virus 3CL_{pro} in complex with NEMO has been described by Ye et al. (ref. 16) as well.

The manuscript is generally well written, however, in parts it could be more concise. Further the nomenclature should be made consistent throughout the manuscript, in particular for NEMO peptide fragments. Most figures showing protein structures are too crowded to clearly convey their message and should be simplified. For example, the surface representations should be displayed in one color only.

A concern is the very moderate peptide cleavage efficiency of NEMO by 3CL_{pro} as reported in Figure 1, reaching only about 12% after 2 h incubation at a NEMO concentration of 13 μ M and 3CL_{pro} concentration of 5 μ M. The authors should at least discuss these numbers and their relevance in physiological conditions.

We thank Reviewer #2 for the thorough analysis of our manuscript. We have responded and revised the manuscript as appropriate to address the several important points raised. Our responses to the Reviewer's comments and questions are written in blue font. In the manuscript, all the changes are tracked and the lines of those related to the reviewer's comments are informed in each response. Please note that, following the Nature Communications formatting instructions, the subheadings in Results are no longer numbered and the subheadings in Discussion were removed.

In particular, we disagree with the Reviewer about the study being partly limited due to the referred studies. Wenzel et al (ref. 19 - the first author name was corrected, it) demonstrate the biological relevance of NEMO cleavage by 3CL_{pro}, which supports the importance of our study in understanding the molecular details underpinning this process. Our study is unique and highly complementary to the aforementioned report in unraveling key structural features of 3CL_{pro}-NEMO binding and predicting their connections with differences in pathogenesis across species. The connection between microvascular pathology and cleavage of NEMO by 3CL_{pro}

demonstrated with *in vitro* and *in vivo* data by Wenzel et al. also help to address the Reviewer's fair concern about the relevance of a moderate peptide cleavage efficiency in physiological conditions. In addition, the authors note that the apparent catalytic efficiency of 3CLpro on cleaving NEMO is in the range that has been reported for self-processing cleavage between nsp4 and nsp5. We added text (lines 148-153) to emphasize the complementarity of Wenzel et al. in the Introduction and as part of the discussion (lines 1183-1185) on the relevance of NEMO cleavage by 3CLpro despite its moderate catalytic efficiency. Finally, our study predicts that even a few non-conserved residues either on 3CLpro or NEMO can significantly affect binding affinity (Section "Predicted NEMO binding affinity differs among host and virus species"). Between PEDV and SARS-CoV-2 3CLpro, there are relevant differences in the binding site (Prates et al., 2021), including residues that form persistent contacts with NEMO in our simulations and predicted hot spots, such as M49 and Q189 (S50 and P158 in PEDV 3CLpro).

Furthermore the reported structure of 3CLpro in complex with NEMO exhibits an unusually large Rfree gap indicating potential model bias. The authors should provide omit maps for the bound NEMO peptides to support their binding hypotheses.

We thank the reviewer for their valuable suggestion. We repeated the refinement with phenix and the R and Rfree are 21.7 and 31.3% for the NEMO-bound and 18.8 and 25.7% for the 3CLpro C145S structure. To assess the possibility of over-refinement, we also performed refinement by cutting the higher resolution data to 2.5 Å. This refinement also resulted in a similar gap between R and Rfree values. Additionally, we have included a composite omit map for the NEMO peptides (Supplementary Figure S3a and S3b).

Some more detailed comments:

65: citation should read "2,3". We have amended the manuscript accordingly (line 95).

91: "form" instead of "from"? We thank the reviewer for pointing this out. We have amended this to reflect that we are referring to 3CLpro from PEDV (line 154).

137: Maybe reference 21 (Lampe et al. The SARS-CoV-2 main protease Mpro causes microvascular brain pathology by cleaving NEMO in brain endothelial cells. Nat. Neurosci. 24, 1522-1533.) should already be cited in the introduction to underline the physiological importance of NEMO cleavage by 3CLpro. This reference is now cited in the Introduction (lines 148-153 - the first author name was corrected to Wenzel).

139: Figure 1, overall low cleavage efficiency. Was a longer incubation attempted and, was 100% cleavage reached? If not, what was the reason? Can the authors exemplarily show the analysis of the raw data (HPLC chromatogram/MS analysis)? In the PAGE experiment (Fig. S2), nearly complete cleavage was observed. In the MS experiment, only the time points shown were taken (Fig. 1b). We have submitted, as Source Data, the raw data for the human NEMO cleavage MS analysis performed in Fig. 1b.

146: Figure 2, the caption does not clearly explain the figure. For example, the NEMO peptide is shown with a mesh. However it is not clearly written what kind of electron density is depicted (2Fo-Fc?), also the contour level as well as the carving radius must be stated. We have amended

the figure legend to contain the facts that, first, the density is an omit map. Secondly, the carving radius is 1.9 Å and the contour level is 1.0 sigma.

167: Please specify the NEMO fragment in a consistent way. We have standardized how the NEMO fragment is specified throughout the text. It is referred to inform species and amino acid range, e.g., hNEMO₂₂₆₋₂₃₅.

169: “Chain B and C are each bound [...] (Figure 2a).” Chain B is not depicted in Figure 2a. We have re-formatted Fig. 2 as well as the text to show chain B bound to NEMO in Fig 2b.

192-198: To which wt structure do the authors compare their complex structure? If this is the wt structure also reported in the manuscript, this should be mentioned. An additional figure to demonstrate the conformational changes would be helpful here. It might also be useful to take advantage of the many wt structures of 3CLpro that are already in the PDB to see if these conformational changes are indeed induced through the peptide or are also observed in other structures.

We thank the reviewer for their suggestion. We have amended the manuscript to indicate that the 1.45 Å WT SARS-CoV-2 3CLpro is our reference structure (line 435). We have added the Supplementary Figure S3c, to show these conformational changes. We have added the Supplementary Table S1, with a comparison to many other WT, and mutant structures (at both room and cryogenic temperatures), bound to a range of different ligands to illustrate how the formation of H-bonds between the substrate and both Thr26 and Thr190 facilitate the described conformational change.

209-210: It would be helpful to show all the mentioned pockets in Figure 2. For this a unicolored surface would be more insightful than the current illustration.

In Fig. 2c and 2d, we have changed the surface colors and added the residue numbers for the hNEMO and C-terminal substrates alongside the P-identities of these residues. This should assist in identifying the pockets, which have been defined in Table 1 and Supplementary Table S5.

228-242: please compare also with other 3CLpro/peptide complexes, for example PDB 7KPH and 7JOY from Lee et al. (ref. 10 in manuscript) and 7N6N (N- and C-terminus bound to 3CLpro) from Noske et al (ref. 29). Are the interactions observed in the author’s structure also seen in these structures? Furthermore the NEMO/3CLpro complex and its binding mode should also be compared to the structure of SARS-CoV 3CLpro with a longer N-terminal substrate (PDB 2Q6G) to derive a more general binding model for peptides to 3CLpro.

We thank the reviewer for their suggestions. We have compared our hNEMO-bound SARS-CoV-2 3CLpro C145S structure to PDB IDs: 7KPH, 7JOY, 7N6N, and 2Q6G in Supplementary Table S5. Additionally, we have extended Table 1 to include a list of the interactions of the SARS-CoV-2 3CLpro C145S substrate-binding groove with the C-terminal tail for comparison with the NEMO-specific interactions. We then discuss the conservation, or lack thereof, of the interactions observed in Supplementary Table S5 in Discussion (1078-1108). This broad comparison revealed highly conserved interactions as well as the plasticity of the 3CLpro substrate-binding groove that underlies the substrate versatility of this enzyme.

246/Suppl. Fig S3: S3a, in inset the figure would become better interpretable when the C-terminus is not shown as surface but only as stick model.

We have moved Fig. S3a to the main text (Figure 2b,e). In Fig. 2e, we depicted chain A as a unicolored pink surface and chain B as a purple cartoon. The C-terminal tail is depicted as sticks juxtaposed on the cartoon and surrounded by an omit map of its density to show how the C-terminal tail fits into the interfacial binding site.

246-253: This section needs to be supported by an additional supplementary figure showing the proposed binding pocket and interactions. This becomes not clear from Fig S3a.

We have added a clearer representation of the interfacial C-terminal binding site (Fig. 2e). This is supplemented by a description of the interactions of the C-terminal tail with the 3CLpro in this binding site (lines 629-671).

258: Please specify the NEMO fragment in a consistent way. We have amended the manuscript to specify the NEMO fragment in a consistent way, specifying the species and the amino acid range (e.g., hNEMO₂₂₆₋₂₃₅).

259 ff: “Arg298 of chain B is closer...”. A supplementary figure would be helpful to illustrate this comparison.

This analysis in the subsection “Interfacial C-terminus may attenuate self-inhibition in 3CLpro dimer”)has been pointed out as unclear by a second reviewer. Without any significant impact to this work, we have decided to remove it, as the positions of the guanidinium N atoms in Arg298 are not well defined by the density in our crystal structure. In that subsection, we focus on the analysis of the predicted relative binding affinities of crystallographic hNEMO peptide in the catalytic site (Fig. 2c) and C-terminus in the interfacial (Fig. 2e) and catalytic binding sites (Fig. 2d). Please see response to the next question from the Reviewer.

274-276: “These comparable binding affinities...”. Is there precedent for this type of double binding site for decreasing self-inhibition? Or is this unique property of SARS-CoV-2 3CLpro? The authors should state that these affinities are predicted or otherwise try to experimentally validate their statement.

The dynamical interplay between active and inhibitory conformations has been reported both for intramolecular self-inhibition (<https://pubs.acs.org/doi/10.1021/acs.jcim.1c00840>) and intermolecular inhibition by homodimerization (<https://www.pnas.org/doi/full/10.1073/pnas.0606603103>). We are not aware of a study reporting a mechanism of regulating inhibition using alternative binding sites, similar to what we proposed for 3CLpro. We have edited the paragraph to clarify that it is a hypothetical mechanism. We have added text to point out that the fact that both conformational states are present in our crystallographic structure of 3CLpro supports this hypothesis, suggesting comparable binding affinities between the two modes *in vitro* (lines 688-753).

291, figure 3: Panel c is missing labels for the x-axis. Also please indicate the active site on 3CLpro to guide orientation of the reader. Furthermore, please explain the discrepancy between the run productive run length mentioned of 112 ns in the Materials and Methods section and the

graph at a run length of 190 ns. Also, please specify the NEMO fragment in a consistent way. The label was added to the x-axis. We also specified in the Methods Sections that the simulations of SARS-CoV-2 3CL_{pro} bound to human and mouse NEMO peptides were extended to 192 ns to compare the stability of these ligands in the catalytic site (please see the response below to the Reviewer's comment about different simulation times, referring to lined 703-705). The NEMO fragments are now specified according to the species and the corresponding amino acid range (e.g., human NEMO aa. 226-235 is noted as hNEMO₂₂₆₋₂₃₅).

361: Visualization of results in Fig. 4b should be improved. Can the results of the MD/ML affinity prediction be quantified in a similar manner to the QM methods? Otherwise give the results for the individual methods in the supplement. Thank you for making this clarifying point. We have updated Fig. 4b to display the predicted rankings for all the machine learning methods used and we have added the Supplementary Table S3 where these predicted binding affinities are presented for each of the different ML methods. The values of the predicted affinities are unitless, as they are expressed as $-\log(K_d)$. It should be noted that the predicted affinities are used for *rankings* and not the *absolute affinities* as they may not be directly comparable with experimental values. The reason for this is that the ML models were trained using a database (PDBbind) that consists largely of proteins bound to relatively small, drug-like ligands. The NEMO peptide is, in general, much larger than the ligands found in PDBbind. This is why the predicted affinities of NEMO binding are incommensurate with the binding affinities of the training data. Nonetheless, the physicochemical space covered by PDBbind is vast and is representative of that found at the NEMO:3CL_{Pro} binding interface. This is now explained in Methods (lines 1542-1550).

In addition, we moved the QM-predicted binding affinities to the Supplementary Table S3. These QM-calculated internal binding energies can be significantly larger than experimental binding free energies due to the lack of entropy contributions. Therefore, similarly to MD/ML predictions, The QM-based binding energies should not be considered an absolute measure of binding affinity but instead a scoring function to rank relative binding affinities (<https://doi.org/10.1002/jcc.24850> and <https://doi.org/10.1002/cphc.201701104>). We added this statement to the manuscript (lines 1513-1517).

379: In addition to the binding of NEMO to 3CL_{pro} mostly governed by their calculated affinity, it is worthwhile to also include a more detailed analysis of the NEMO coiled coil structure of the template. For example, TWISTER analysis (Strelkov & Burkhardt 2002, PMID 12064933) of PDB model 6MI3 suggests a coiled-coil anomaly just prior to this unfolded region, which could be advantageous for CC dissociation. This increases the interhelical distances and thus destabilizes the hydrophobic core of the coiled coil. This might be worth discussing to better argue why this might be a preferred site for initial unwinding and subsequent 3CL_{pro} binding and cleavage

Thank you very much for this valuable suggestion. We have incorporated the TWISTER analysis to our Results and Discussion Sections complemented by PAIRCOIL predictions of coiled-coil regions, as well as the prediction of molecular recognition features (MoRFs) that undergo disorder-to-order transition upon binding on NEMO (lines 1063-1073, 1211-1232, and Fig. 3c). In short, we hypothesize that the combination between a lower propensity for coiled-coil formation and the presence of MoRFs at this unfolded region of NEMO contributes to the

preferential cleavage at site Gln231 by 3CLpro. For the coiled-coil analysis, we covered different regions of NEMO using the PDB structures 6MI3 (aa. 51-112), 3CL3 (aa. 198-249), and 6YEK (aa. 259-343).

402: Figure 5, black line seems of lower resolution than rest of image. We have improved the visual quality of the line.

415: Please state you compare a structure solved at cryo conditions with a structure solved at room-temperature.

Jaskolski et al., 2021 (<https://doi.org/10.1107/S2052252521001159>) indicates that there are no significant structural changes between crystal structures solved at room temperature and cryogenic 3CLpro structures; We now mention that we are comparing our cryo-structure to a room-temperature crystal structure (PDB_ID: 7N89) (lines 1032-1033).

703-705: Could the authors clarify why different run length parameters were used for the different 3CLpro and NEMO complexes. All the unrestrained production runs were preceded by long equilibration phases and performed during 112 ns, except for the simulations of SARS-CoV-2 3CLpro bound to mouse and human NEMO₂₂₇₋₂₃₄, which were extended to 192 ns. The two reason for that are fairly minor: First, we wanted to verify if longer simulations would capture the detachment of mouse NEMO₂₂₇₋₂₃₄ from the catalytic site, as observed early in our simulations of the long construct of mouse NEMO and, second, because we prioritized using extra computing hours granted for this project with simulations of the crystal structure obtained in this study. For improved clarity, similarly to what was done for MD/ML simulations (Supplementary Table S8), we have added the Supplementary Table S9, which summarizes the classical MD simulations of 3CLpro bound to NEMO used for interspecies comparisons. The restrained molecular dynamics simulations used in the MD/ML approach, combined with machine learning models, served the particular goal of predicting relative binding affinities of protein-ligand complexes and, therefore, involved a different schema of simulation phases. This method was derived from a MD-based protocol of structure refinement (Heo et al., 2021).

Supplement:

89: Please explain the different coloring of the table text in the table legend. We have updated the Table legend, defining the coloring code as follows: “Red font indicates the P1 site (Gln231 in human NEMO, hNEMO), light green highlights the reference sequence, and light purple, species harboring a different motif.”

100: Resolution limits of outer shells are missing. Please also state the overall I/sigma and not only I/sigma for the outer shell. We have added the resolution limit for the outer shell and overall I/sigma to Supplementary Table S7.

112: Figure S1, please also state the identity between mouse and human NEMO to highlight their identity. We have updated the figure legend by stating that there is 87% identity between human and mouse NEMO.

117: Figure S2 would be more intuitive if the cleavage products of the different NEMO constructs were directly labeled.

We thank the reviewer for their suggestion and have done so in the Supplementary Figure S2b.

122: Figure S3a, removal of the surface representation of the C-terminal peptide would make the figure clearer. Also, state the type of electron-density depicted in panel b and c as well as their contour level.

We have amended this figure accordingly. Fig S3a has been moved to Fig 2. We have also moved the inset figure of the interfacial site to an entirely new panel (Fig. 2e) to facilitate interpretation. Electron densities have been updated with information about the type of electron density map (omit maps) as well as the contour level and carve radius used.

134: Figure S4, a scheme as additional panel showing the position of analyzed interchain distances would enhance interpretability of MD results.

As discussed in the response above (259ff), without any significant impact to this work, we have decided to remove this analysis in the subsection “Interfacial C-terminus may attenuate self-inhibition in 3CLpro dimer” (line 684) and, consequently, this supplementary figure.

147: Figure S6, please also show same analysis for SARS-CoV 3CLpro to highlight differences mentioned in the text.

Panel b, showing this analysis for SARS-CoV 3CLpro, has been added to this figure (now, Figure S4b). In addition, to explore the impact of the initial conformation in these results, we have added a comment about essential Ca cross-correlation analysis performed using a different structure of SARS-CoV-2 3CLpro as the starting point of our simulations (lines 896-899). The new results also seem to reflect a tighter dimeric packing of SARS-CoV-2 3CLpro compared to SARS-CoV 3CLpro.

Reviewer #3 (Remarks to the Author):

The manuscript characterizes the cleavage of fragments of the NF- κ B Essential Modulator (NEMO) by the 3CLpro protease of SARS-CoV-2. The authors validate the cleavage at one of the previously reported cleavage sites, Gln231, and solve the structure of a short NEMO peptide (226-235) bound to the active site mutant C145S of 3CLpro. The complex structure confirms binding pocket characteristics that modulate substrate selectivity. The authors also propose a role for the C-terminal tails of 3CLpro in quaternary structure interactions and self-inhibition. Machine learning- and physics-based computational methods are applied to compare the interaction of NEMO with SARS-CoV-2 and other coronaviruses and its effect on pathogenesis.

The manuscript is well crafted, the methodology is described in detail and the conclusions are in line with the data presented.

We thank the reviewer for the positive feedback and for the thorough analysis of our study. The points raised for clarification were very helpful to improve this manuscript. Our responses to the Reviewer’s comments and questions are written in blue font. In the manuscript, all the changes are tracked and the lines of those related to the reviewer’s comments are informed in each

response. Please note that, following the Nature Communications formatting instructions, the subheadings in Results are no longer numbered and the subheadings in Discussion were removed.

Please find detailed below some of the concerns with the manuscript.

Results: 2.1. The cleavage of hNEMO at different sites by SARS-COV-2 3CLpro was previously reported (manuscript ref. 20), as well as the catalytic efficiency of the cleavage at Gln231, which affect the novelty of these findings. The manuscript could benefit from a comment on the other cleavage sites (Q83, Q205, Q304 and Q313) that were not observed in this study. The sensitivity of SDS-PAGE detection may be an issue for low % cleavage.

These experiments were intended to reproduce results from other laboratories when those were only available as preprint (<https://doi.org/10.21203/rs.3.rs-86988/v1>). Our major focus in this study was not to prove cleavage, but to explore and describe the molecular basis of this process. Minor cleavage products would require proteomics analysis, which we considered beyond the scope of a primarily structural and computational study.

A comment on the preferential cleavage at site Q231 relative to the other cleavage sites was added (see lines 235-237). We have further discussed the potential reasons for that using analyses of the coiled-coil structure of NEMO and predictions of molecular recognition features (MoRFs) on NEMO that undergo disorder-to-order transition upon binding. In short, we hypothesize that the combination between a lower propensity for coiled-coil formation and the presence of MoRFs at this unfolded region of NEMO contributes to the preferential cleavage at site Gln231 by 3CLpro indicated by our cleavage assays (Supplementary Fig. S2). Please see lines 1063-1073, 1211-1232, and Fig. 3c.

Results 2.2-2.7. As the authors mention, the Protein data Bank is replete with structures of SARS-CoV-2, including WT apo protein, complexes with peptides, peptidomimetics and small molecule inhibitors, and C145A/S mutants. The complex of a similar NEMO peptide with 3CLpro for Porcine Epidemic Diarrhea virus was also reported (ref. 14). The complex structure in this manuscript highlights the characteristics of the SARS-CoV-2 3CLpro binding site as it interacts with NEMO. The additional insight may further our understanding of the selectivity/versatility of 3CLpro. The structure also details the binding of the C-terminal tail to the substrate binding site and at dimers interfaces. Similar findings were reported previously. The description of the C-terminus binding at the dimers interface (2.7) could benefit from a figure, it is detailed but difficult to follow. Supplementary Figure S3 does not show all the described interactions and the two chains are depicted in similar shades of pink.

We have moved Supplementary Figure S3a to the main text (Figure 2b and e) to make it easier to reference and changed the depictions of various parts of the figure. Chain B is no longer represented as a surface). The C-terminus has been surrounded by its omit map electron density (with listed carve radius and contour level), to show how it interlocks with the dimer interface and the residues that are involved in interactions with the C-terminus have been listed. We have also changed the color scheme of Fig. 2e to make it easier to interpret (chain A is light pink and chain B is deep purple).

The statistics from the structure refinement indicate $R_w/R_f = 0.214/0.309$. The R_{free} value is high for a 2.14 Å dataset, and the large spread between R_w and R_f may indicate that the structures were over refined. The authors should try an alternative refinement strategy. The same is valid for the 3CLpro C145S structure ($R_w/R_f = 0.194/0.273$).

We thank the reviewer for their comment and valuable suggestion. We repeated the refinement with phenix and the R and R_{free} are 21.7 and 31.3% for the hNEMO-bound and 18.8 and 25.7% for the C145S structure, respectively. To assess the possibility of an over-refinement, we also performed refinement by cutting the higher resolution data to 2.5 Å. This refinement gave a similar R and R_{free} gap (Supplementary Table S7).

Results 2.8-2.9. The molecular dynamic simulations examine the interaction of a longer NEMO fragment (190-270) with 3CLpro. The model for the long NEMO fragment was built using the NEMO structure from PDB 6MI3 as a template (the model also appears to be used in the analysis of hydrophobic interactions within the NEMO dimer, discussion). 6MI3 reports the structure of a different domain of NEMO (51-112) fused to long N- and C-terminal ideal coiled-coil adapters. Both portions seem inadequate: the GCN4-derived portion and the NEMO portion, which due to the low coiled-coil propensity in the central sequence adopts a peculiar structure. The crystal structure of NEMO(150-272) was reported (in complex with a viral FLIP) and may be a better starting point for the simulations (3CL3). Also the manuscript would benefit from more detail on how the “partially unfolded” NEMO(190-270) structure was generated and the comparison with dimeric NEMO.

We thank the Reviewer for this thorough analysis of our Methods and for this comment. We agree that PDB 6MI3 is not an ideal starting point for the simulations. The crystal structure of NEMO (aa. 150-272) could have been a better choice, but as NEMO interacts with FLIP, it may not either well represent the state preceding its binding to 3CLpro.

Our simulations of NEMO(190-270)-bound WT 3CLpro reproduced interactions captured experimentally and suggested that the competing interactions within the NEMO dimer and those with 3CLpro can affect binding of mouse and human NEMO to 3CLpro differently. Given the limitation of obtaining a more robust model for these simulations, we opted to use a different approach to explore the effects of coiled-coil propensity on 3CLpro-NEMO binding. Please see lines 1063-1073, 1211-1232, and Fig. 3c, and the summary below.

Analysis of coiled-coil parameters was performed using TWISTER (Strelkov and Burkhard 2002), complemented by predictions using PAIRCOIL (Berger et al. 1995). ANCHOR (Dosztányi et al. 2009; Oates et al. 2013) was used to predict potential disordered binding regions, or molecular recognition features (MoRFs), in NEMO. For the coiled-coil analysis, we covered different regions of NEMO using the PDB structures 6MI3 (aa. 51-112), 3CL3 (aa. 198-249), and 6YEK (aa. 259-343). We hypothesize that the combination between a lower propensity for coiled-coil formation and the presence of MoRFs at the unfolded region of NEMO contributes to the preferential cleavage at site Gln231 by 3CLpro compared to the other sites. In addition, we predict that mouse NEMO has higher coiled-coil propensity near the Gln231 compared to human NEMO (Fig. 3c). Because of that, we hypothesize that competing interactions within the NEMO dimer may more effectively destabilize mouse NEMO than human NEMO, which appears to be more strongly bound to the catalytic site of 3CLpro in our simulations with NEMO peptide-bound 3CLpro (Fig. 3e). These analyses enriched the

discussion without changing our previous predictions, so they replaced the previous analysis of the simulations of 3CLpro bound to the long construct of NEMO. Still, we note that the precision of our coiled-coil and MoRF analyses are limited to the use of predictors and to the use of NEMO structures that may not accurately represent the state preceding its binding to 3CLpro, i.e., in PDB id 3CL3, NEMO interacts with other proteins and in PDB id 6MI3 it is fused to N- and C-terminal ideal coiled-coil adapters. Therefore, further studies are warranted to test our hypotheses.

The differences reported in 2.9 lines 325-327, comparing mNEMO and hNEMO, seem to be within the reported errors: the authors should add a comment on why they are relevant or remove. Small differences in interaction profile and molecular rigidity observed in molecular dynamics simulations of intrinsically disordered proteins were demonstrated to predict significant differences in proteolytic efficiency (Prates et al., Chem Sci, 2018, DOI: 10.1039/C7SC05016J) (lines 1112-1115). We have added to the Concluding Remarks a sentence stating that our results reinforce the importance of performing site-directed mutagenesis and enzymatic assays to quantify the contribution of non-conserved amino acids in 3CLpro and NEMO across different species to proteolytic efficiency (lines 1195-1209).

Simulations utilizing the experimental complex structure of this work and comparative analysis of betacoronaviruses: a comment on the impact that starting from an experimental complex structure (NEMO+ SARS-CoV-2) versus a model (NEMO + other coronaviruses 3CLpro) may have on the predicted binding affinity may be helpful.

We have added text to point out that the impact of using a crystal structure only for the NEMO peptide bound SARS-CoV-2 3CLpro, and not for the other complexes, is not fully clear (lines 921-922). However, we describe evidence that suggests the robustness of our predictions and encourage future binding assays to evaluate them. For example, the highest predicted binding affinity of the NEMO peptide to SARS-CoV-2 3CLpro could be thought as a biased result from using a crystal structure as the starting point for this case. However, we find that energy minimized-only structures, which are much less perturbed than those used in the MD/ML protocol, do not provide consistent results as our methods do and do not confirm this bias, with SARS-CoV-2 3CLpro presenting lower relative binding affinity to the hNEMO peptide compared to other coronaviruses 3CLpro (lines 922-929). Supplementary Table S4 was added to show the predictions for energy minimized-only structures. In addition, these structures have appreciably higher energy of interaction computed using the CHARMM36 force field in GROMACS than the MD-derived structures, suggesting that the MD-based protocol of conformation selection is effective to account for the conformational flexibility of the receptor (lines 1539-1542).

Figure 5 is difficult to read due to the coloring of the residues (same N, O and S) -and possibly the lack of a substrate.

We have changed the background color of the SARS-CoV-2 3CLpro C145S chain C surface to light blue only, and have also changed the colors of the 7N89 residues. While the carbon atoms are still green, different (lighter) color tones are used for oxygen, sulfur, and nitrogen atoms. This should improve the comparison. Furthermore, we have added an additional panel, Fig. 5b, a

juxtaposition of the hNEMO and N-terminal peptide substrates to show the conserved pose of the two substrates in the substrate-binding groove.

Methods:

Line 613: “25 ug of TEV per ug of 3CLpro”, correct “u” for “μ” or “m”?

We have amended the manuscript to correct this to μ (line 1353).

Lines 621-627: in two different paragraphs two different descriptions of how the peptide/3CLpro crystals were prepared.

We have reordered some paragraphs in this section of the methods to correct this. Specifically, we have used the subheadings “3CLpro WT expression, purification, and crystallization” (line 1313); “3CLpro C145S expression, purification, and crystallization” (line 1343) and “3CLpro C145S-NEMO co-crystallization” (line 1365).

Also:

Line 91: “The structure of 3CLpro from in complex” is not clear

We have amended the manuscript to reflect that we are referring to the structure of PEDV 3CLpro bound to a NEMO heptapeptide (line 154).

Lines 392-393: “non covalent associations of the IKKα/β domains of NEMO” is not clear what the authors mean.

We have amended the manuscript to reflect that hNEMO homodimers associate with neighboring hNEMO homodimers using their IKKα/β domains as the point of interaction (lines 987-991).

Lines 442-443: one is meant to be SARS-CoV rather than SARS-CoV-2? Yes we meant SARS-CoV, we have corrected it (line 1112).

Supplementary Table S3: superscripts a and b are used in the table but not explained. We have defined superscript a and b underneath Supplementary Table S7.

Supplementary Figure S2: the expected cleavage band from NEMO(232-250) is not observed in lane 6 (it is observed elsewhere). The band is likely weak because of the small size and the lane may have a lower loading than the others?

We thank Reviewer #3 for the comment on the lack of the band corresponding to NEMO (232-250) in lane 6. It is possible that we loaded a smaller amount of sample in this lane compared to the others. However, we feel confident that the cleavage did happen at Q231 since we do see a clear band at the molecular weight 42.59 kDa corresponding to GST-96-231, larger of the reactants MW components of the cleavage reaction. Supplementary Figure S2 has been updated accordingly with MW estimates of cleavage fragments (*see below*). We also added a table (Supplementary Figure S2b) with the MW calculations of all the combinations of cleavages at Q205, Q231, Q304 and Q313, and highlighted the MW combinations visible in the gel pattern in green.

a

b

Lanes	96-250		221-250		GST-96-250		GST-221-250		GST-221-339	
3CLPro addition	(+)	(-)	(+)	(-)	(+)	(-)	(+)	(-)	(+)	(-)
Uncleaved (-)	17.87	17.87		3.59	44.85	44.85	30.54	30.5	40.85	40.85
Q205	12.70 + 5.19		N/A		39.68 + 5.19		N/A		N/A	
Q231	15.61 + 2.28		1.33 + 2.28		42.59 + 2.28		28.31 + 2.28		28.31 + 12.56	
Q304	N/A		N/A		N/A		N/A		36.75 + 4.25	
Q313	N/A		N/A		N/A		N/A		37.80 + 3.19	
Q205 & Q231	12.70 + 2.93 + 2.28		N/A		39.68 + 2.93 + 2.28		N/A		N/A	
Q231 & Q304	N/A		N/A		N/A		N/A		28.31 + 8.46 + 4.25	
Q304 & Q313	N/A		N/A		N/A		N/A		36.75 + 1.07 + 3.19	
Q231 & Q313	N/A		N/A		N/A		N/A		28.31 + 9.51 + 3.19	
Q231 + Q304 + Q313	N/A		N/A		N/A		N/A		28.31 + 8.46 + 1.07 + 3.19	

Reviewers' Comments:

Reviewer #1:

Remarks to the Author:

The authors have done a really nice job addressing all of the Reviewers' comments, which called for extensive changes in some places. The manuscript has been significantly strengthened, and I believe it now warrants publication.

Reviewer #2:

Remarks to the Author:

The authors have addressed the majority of our comments and concerns. This has significantly improved the quality of the manuscript. However, a few concerns remain, which are mentioned below. Provided these can be resolved in a minor revision, we suggest acceptance of the manuscript.

A main concern is the overall poor quality of the X-ray structure of the NEMO/3CLproC145S complex and the corresponding apo structure. We strongly recommend to try to optimize data processing and analysis and, if this is not improving the data quality, to consider repeating the diffraction experiment. The poor quality is reflected in the high R free gap for both structures and the poor stereochemistry (Ramachandran outliers). During the refinement step, the authors could try different weighting schemes (X-ray/stereochemistry and X-ray/ADP) in phenix.refine. The high number of Ramachandran outliers in C145S 3CLpro structure might be an indication that tighter stereochemistry restraints could improve the refinement. Regaining the same large Rfree gap with another refinement program does not exclude the possibility of over-refinement. (It rather stresses the point that this is not an artifact of the refinement program, but rather the data or the model.) Did the crystals in the P1 space group suffer from radiation damage? Can omission of diffraction images improve the rather poor merging statistics for both 3CLpro C145S structure (and subsequently improve refinement)?

The authors respond to our concern of the comparatively low cleavage efficiency with referring to the similarly low efficiency in the cleavage of nsp4/5 that has been reported in the literature. Please add a suitable reference. We cannot find this mentioned in the manuscript. Also, it might be worthwhile to discuss potential additional factors that might influence the cleavage in the cell. Interestingly, in contrast to the low cleavage efficiency in the HPLC-MS assay with hNEMO215-247 (Figure 1), the cleavage of mNEMO221-250 in the PAGE assay (Figure S1) appears to have reached completion within a reaction time of only 20 min. Can you explain the reasons for this discrepancy (difference in reactions conditions (25 vs 37C, concentrations, sequence differences)?

Further minor comments

Supplement:

Table S1: Thank you for adding this valuable comparison. However, the authors should state the reason for selecting this particular subset of eight structures of the currently >500 SARS-CoV-2 structures available in the PDB for comparison.

Table S5: This is a valuable extension of the manuscript but needs more explanations: What does W mean? Water replacing a substrate interaction? Or is the H bond replaced by an indirect water-mediated hydrogen bridge from 3CLpro to the substrate? What does the strikethrough of some residues mean? It will be easier to read the table if the authors add a column with the interactions observed in the hNEMO/3CLpro structure for comparison.

Table S7: please format consistently. For example, all values for the outer shell in parantheses. Give value for I/σ and outer shell on same line, rather than giving outer shell value together with Rsym.

Figure S1: Please clarify that the numbering of cleavage sites corresponds to human NEMO. Murine NEMO numbering of cleavage sites corresponding to human NEMO Q304 and Q313 differs due to an

insertion.

The protocol for the mNEMO cleavage analysis by PAGE differs between method section and legend of Figure S1: Concentrations of substrate (0.2 vs. 0.5 mg/mL) and enzyme (0.5 vs. 0.25 μ M). Please clarify.

Figure S3: Is the depicted electron density a composite omit map or just an omit map? The information in the rebuttal letter and the figure legend are not clear. Also, the sequence of subpanels and legend text are not fitting.

Subpanel S3c (distance Thr26/Thr190): For which of the four molecules in the asymmetric unit of C145S and C145S/NEMO is the distance shown? How is this in the other molecules in the asymmetric unit? Or give reason for selecting only this particular one.

Paragraph "Thr26 and Thr190 of 3CLpro pin the NEMO peptide into 3CLpro": Please give both measured distances for each Thr26/Thr191 pair in the C145S structures also in the manuscript main text and not only in the Table S1.

Methods:

I 1307: Was quenching done at 37C or at a higher temperature?

Main text:

Please check references to supplementary tables after insertion of addition table S1. For example line 787 should probably refer to Supplementary Table S2.

Reviewer #3:

Remarks to the Author:

Major:

The difference between Rwork and Rfree values for structures 3CLPro-NEMO and 3CLPro C145S is still unacceptably high (see slider graphic of percentage score in the PDB report).

Since coordinates and electron density data are not released yet, this analysis is based only on the PDB validation report.

This is clearly reported in the statistics calculated in the PDB validation reports in comparison with all PDB structures as well as with comparable resolution structures. The literature references dealing with the Rwork/Rfree spread abound and I am sure are well known to all.

Protein Crystallography and Drug Discovery Jean-Michel Rondeau, Herman Schreuder, in The Practice of Medicinal Chemistry (Fourth Edition), 2015

"Since refinement programs aim at minimizing the difference between observed and calculated amplitudes (hence the R-factor), an unbiased indicator is needed to monitor the progress of refinement. Brunger proposed excluding a subset of reflections from refinement and using these reflections only for the calculation of a "free" R-factor [79]. If refinement is progressing correctly, the free R-factor will drop as well"

"The risk of overfitting the data has been considerably reduced when the concept of cross-validation was introduced [52]. This procedure involves setting aside a certain proportion of the data, the test set (ca. 5–10% of the data), which does not enter refinement, but is only taken for quality control. In this way, an independent quality indicator is generated that has a high correlation with the overall phase error and thus with the accuracy of the structural model. The free R-factor (Rfree) should remain as close as possible to the conventional crystallographic R-factor during refinement. Usually, the two values stay within 2–5% of each other if refinement proceeds well, with Rcryst having final values of 18–25% and Rfree of 22–30%. The lower the value, the better is the match between experimental data and structural model."

In the most straightforward interpretation such a large spread between Rw and Rf indicates overfitting of the data, resulting in a poor fit of the model to the actual experimental data. Many suggestions are available in the literature on how to overcome such issues, including extensive manual rebuilding of

the model, use of NCS if present, adjustment of ADP, TLS, B factors, optimization of refinement weights. Dataset/sample specific issues may affect statistics and should be discussed if statistics deviating largely from acceptable standards are reported.

Specifically for the two models:

3CLPro-NEMO solved at 2.14 Å; Rw/Rf difference is 9.5%

The Rfree value and Rw/Rf gap indicate poor fit of the model to the data. The "slider" graphic of the PDB report shows very poor statistics for Rfree vs all PDB structures as well as vs comparable resolution structures (actually almost the worst possible score).

Large stretches of residues are reported with poor fit to the electron density (especially chain C and D).

3CLPro C145S solved at 2.45Å; Rw/ Rf difference 7.9%. This is still high. Chain B and C show a number of residues with poor fit to the electron density. The B factors are high, this could benefit from a discussion in the manuscript.

Minor:

-Figure 3b: The authors report they have removed the simulation of NEMO(190-270)-bound WT 3CLpro, in favor of the analysis of coiled-coil parameters. This is acceptable, but figure 3b needs to be removed as the manuscript contains no report on how the structure was generated and it could be misleading.

-Figure 3c is very difficult to understand, due to the size, lack of labeling and very succinct description in the legend. The authors should clearly indicate what values in the graphs prompts them to state "suggests a coiled-coil interruption at Tyr241, adjacent to the unfolded binding core of hNEMO," (lines 831-832). The evidence of an "unfolded binding core of hNEMO" needs to be strengthened and more clearly explained; at this stage it would be better stated as a hypothesis.

-It seems like in different parts of the manuscript a reader could object that a circular argument is used to justify the computational strategies utilized (unfolded binding core of NEMO and comparative analysis of betacoronaviruses). If you allow the over-simplification of the argument for brevity it may seem that the authors state in their rebuttal that the experimental results are validated by the computational results and the computational methods are valid because they reproduce the experimental results. If the authors could provide a more objective validation of their choice of computational methods the manuscript would be improved.

-990: "the IKKa/b domains of ne NEMO homodimer: do the authors mean the IKKa/b-binding domains of NEMO?"

-990 interacts should read interact

REVIEWER COMMENTS / Authors' responses in blue font

Reviewer #1 (Remarks to the Author):

The authors have done a really nice job addressing all of the Reviewers' comments, which called for extensive changes in some places. The manuscript has been significantly strengthened, and I believe it now warrants publication.

We appreciate the Reviewer's comments and suggestions, which truly helped improve the manuscript and strengthen this study.

Reviewer #2 (Remarks to the Author):

The authors have addressed the majority of our comments and concerns. This has significantly improved the quality of the manuscript. However, a few concerns remain, which are mentioned below. Provided these can be resolved in a minor revision, we suggest acceptance of the manuscript.

Once more, we thank the reviewer for the thorough analyses of our manuscript, for their valuable suggestions, and for the positive feedback. We hope we have satisfactorily addressed their remaining requests.

Our responses to the Reviewer's comments and questions are written in blue font. In the manuscript, all the changes are tracked and the lines of those related to the reviewer's comments are informed in each response.

A main concern is the overall poor quality of the X-ray structure of the NEMO/3CLproC145S complex and the corresponding apo structure. We strongly recommend to try to optimize data processing and analysis and, if this is not improving the data quality, to consider

repeating the diffraction experiment. The poor quality is reflected in the high R free gap for both structures and the poor stereochemistry (Ramachandran outliers). During the refinement step, the authors could try different weighting schemes (X-ray/stereochemistry and X-ray/ADP) in phenix.refine. The high number of Ramachandran outliers in C145S 3CLpro structure might be an indication that tighter stereochemistry restraints could improve the refinement. Regaining the same large Rfree gap with another refinement program does not exclude the possibility of over-refinement. (It rather stresses the point that this is not an artifact of the refinement program, but rather the data or the model.) Did the crystals in the P1 space group suffer from radiation damage? Can omission of diffraction images improve the rather poor merging statistics for both 3CLpro C145S structure (and subsequently improve refinement)?

We thank the reviewer for their insightful guidance regarding the crystallographic data quality. We have repeated the data processing employing some of the techniques suggested by the reviewer. Because of the triclinic space group, we had collected multiple data sets/runs with different detector distances to improve the completeness at higher resolution (wider scattering angles) and along the seams of the detector modules. Since we were collecting data from different parts of the crystal, the crystal may not have suffered from radiation damage. We inspected the images and removed lesser quality images of some runs while paying attention to merging statistics. The weighting schemes were also adjusted during the refinements. This improved the Rfree and Rwork refinement statistics and the gap between them, too, while simultaneously improving the fitting of the model to the map, hence the smaller number of Ramachandran outliers. This resulted in Rwork/Rfree values for the C145S-Nemo structure of 30.0/24.5 compared to the previous 30.9/21.4 with 5 Ramachandran outliers, previously 6. The Rwork/Rfree values for the C145S structure are 19.6/23.7 with 6 Ramachandran outliers improved from 19.4/27.3 and 18 respectively. These changes have been amended in Supplementary Table S7. Details of the new refinement method used have been added to the Methods section of the manuscript (lines 913-922).

The authors respond to our concern of the comparatively low cleavage efficiency with referring to the similarly low efficiency in the cleavage of nsp4/5 that has been reported in the literature. Please add a suitable reference. Also, it might be worthwhile to discuss potential additional factors that might influence the cleavage in the cell. Interestingly, in contrast to the low cleavage efficiency in the HPLC-MS assay with hNEMO215-247 (Figure 1), the cleavage of mNEMO221-250 in the PAGE assay (Figure S1) appears to have reached completion within a reaction time of only 20 min. Can you explain the reasons for this discrepancy (difference in reactions conditions (25 vs 37C, concentrations, sequence differences)?

The appropriate references were added (Wenzel et al., 2021 and Krichel et al., 2020; lines 747). We have added text (lines 745-749) and a valuable reference that demonstrates that reaction rates *in vitro* cannot be simply extrapolated to the cell environment (Zotter et al. JBC, 2017, DOI:10.1074/jbc.M117.792119). We decided to not prolong this discussion to respect the word limit of the journal.

Our simulations indicate that the single substitution V232A in NEMO from human to mouse perturbs binding to 3CLpro. However, systematic experiments are essential to determine how this impacts cleavage efficiency. The primary goal with the two experiments was not this comparative analysis, but the reviewer brings an important observation. It is possible that the faster cleavage of mNEMO relative to hNEMO peptide is authentic, even though the higher temperature in the PAGE experiment has to be accounted for as well. We have added text to the manuscript to point this out (lines 728-732).

Further minor comments

Supplement:

Table S1: Thank you for adding this valuable comparison. However, the authors should state the reason for selecting this particular subset of eight structures of the currently >500 SARS-CoV-2 structures available in the PDB for comparison.

7T2T, 7T2U and 7T2V are the structures solved in this study. These structures were compared to both cryogenic- and room-temperature crystallography structures that were

solved in complex with substrate peptides to assess what combination of hydrogen bonds formed between Thr26/Thr190 and the substrate peptide causes a decrease in distance between these two residues. We addressed this by selecting, for the cryogenic condition, a peptide-bound C145S 3CLpro mutant (7N6N) and both WT and C145S 3CLpro structures without bound peptide substrates (7KPH and 7N5Z, respectively). For room temperature conditions, a peptide-free WT structure (7JUN) and a C145A 3CLpro structure bound to an N-terminal peptide (7N89) were chosen. There is no available room temperature structure of peptide-free C145A 3CLpro for comparison. Room-temperature inhibitor-bound 3CLpro structures (telaprevir/boceprevir-bound WT 3CLpro) were chosen to demonstrate that ligand-binding in the active site groove without H-bonding to both Thr26 and Thr190 does not cause the decrease in distance observed in structures where these H-bond interactions are formed. We added these explanations about the selected structure choices in the legend of Supplementary Table S1.

Table S5: This is a valuable extension of the manuscript but needs more explanations:

What does W mean? Water replacing a substrate interaction? Or is the H bond replaced by an indirect water-mediated hydrogen bridge from 3CLpro to the substrate? What does the strikethrough of some residues mean? It will be easier to read the table if the authors add a column with the interactions observed in the hNEMO/3CLpro structure for comparison.

“W” refers to water molecules that form direct hydrogen bonds with the substrate, in addition to those interactions formed with 3CLpro. There are also many water molecules that form indirect, water-mediated hydrogen bonds between 3CLpro and the substrate, and those are now indicated with an asterisk in the Table. The strikethrough of residues correspond to those that interact via hydrogen bond or hydrophobic contact with the substrate in the reference structure (hNEMO peptide bound SARS-COV2 3CLpro) and do not make the same interaction in the structure in the comparison set. We added a sentence to explain this in the Supplementary Table S5 legend (lines 210-212).

Table S7: please format consistently. For example, all values for the outer shell in parentheses. Give value for l/σ and outer shell on same line, rather than giving outer shell value together with R_{sym} .

We have made the requested adjustments in Supplementary Table S7. Values for the outer shell are separated from the overall values using parentheses rather than a forward slash. This has been done consistently throughout the table. We have also given the values for mean $\langle I/\sigma(I) \rangle$ and outer shell on the same line, too, rather than giving it with Rsym.

Figure S1: Please clarify that the numbering of cleavage sites corresponds to human NEMO. Murine NEMO numbering of cleavage sites corresponding to human NEMO Q304 and Q313 differs due to an insertion.

Thank you for pointing this out. We have clarified in the figure legend that the residue numbering in the alignment corresponds to the human NEMO sequence and listed their corresponding residues in mouse NEMO (lines 487-488).

The protocol for the mNEMO cleavage analysis by PAGE differs between method section and legend of Figure S1: Concentrations of substrate (0.2 vs. 0.5 mg/mL) and enzyme (0.5 vs. 0.25 μ M). Please clarify.

The enzyme stock solution was 0.5 μ M and 5 μ L were used in a 10 μ L reaction for a final concentration of 0.25 μ M. The substrate stock solutions were 1.4-2.87 mg/mL and diluted in assay buffer to 0.2 mg/mL, and 5 μ L of those stocks were used in a 10 μ L reaction for a final concentration of 0.1 mg/mL. Figure S2 legend was corrected to “a) SDS-PAGE following incubation of five truncations of mouse NEMO (0.1 mg/mL)”. The Methods Section is correct but we added a sentence to state the final reaction concentrations of enzyme and substrate (lines 848-849).

Figure S3: Is the depicted electron density a composite omit map or just an omit map? The information in the rebuttal letter and the figure legend are not clear. Also, the sequence of subpanels and legend text are not fitting.

The density depicted is a composite omit map from Phenix. Amendments to the figure legend have been made to more clearly explain each panel of this figure. For example, we added a more thorough description of how both 3CLpro and the hNEMO peptide are depicted in Supplementary Figures S3a and S3b. Additionally, the figure legend has been amended to reflect that a comparison is drawn between our hNEMO-free SARS-CoV-2 3CLpro C145S structure (PDB_ID: 7T2V), hNEMO-bound SARS-CoV-2 3CLpro C145S (PDB_ID: 7T2U), and the SARS-CoV-2 WT 3CLpro (PDB_ID: 7T2T) structures solved in this study. In our response letter, it was unclear which SARS-CoV-2 WT structure is being analyzed, and we hope that these adjustments clarify this. Furthermore, the subpanel legend text has been rearranged. Whereas, previously, the figure showing the distances between residues Thr26 and Thr190 was erroneously labeled as panel 'a,' it is now labeled as panel 'c.' Consistently, the figures of the hNEMO peptide bound into 3CLpro C145S chains B and C have been re-labeled as panels, 'a,' and 'b,' respectively.

Subpanel S3c (distance Thr26/Thr190): For which of the four molecules in the asymmetric unit of C145S and C145S/NEMO is the distance shown? How is this in the other molecules in the asymmetric unit? Or give reason for selecting only this particular one.

In order to provide a direct visual comparison with the hNEMO-bound C145S schematic in Figure 2c, we used only chain C in the SARS-CoV-2 3CLpro C145S hNEMO-bound structure in Subpanel S3c. We directly compared this to chain C of 3CLpro C145S without hNEMO peptide. Furthermore, because of the space group of our solved 3CLpro WT structure, there is only one chain present in the asymmetric unit for comparison. However, in Supplementary Table S1, we included distances from all chains in the asymmetric units of all structures cited. Specifically, the corresponding distance in chain B of the SARS-CoV-2 3CLpro C145S NEMO-bound structure is 20.6 Å. The manuscript text has also been updated to reflect the distance in each chain (Lines 278-282).

Paragraph "Thr26 and Thr190 of 3CLpro pin the NEMO peptide into 3CLpro": Please give both measured distances for each Thr26/Thr191 pair in the C145S structures also in the manuscript main text and not only in the Table S1.

We have made the necessary amendments to the manuscript by adding the measured distances for the Thr26/Thr190 pair in each chain in the asymmetric unit of both 3CLpro C145S without NEMO and 3CLpro with NEMO (Lines 278-282).

Methods:

I 1307: Was quenching done at 37C or at a higher temperature?

Quenching was performed at 37 °C. The mechanism of quench is LDS denaturation rather than thermal denaturation.

Main text:

Please check references to supplementary tables after insertion of addition table S1. For example line 787 should probably refer to Supplementary Table S2.

Thank you for pointing this out. We have corrected that (line 449) and checked all references in the supplementary tables.

Reviewer #3 (Remarks to the Author):

We again thank the Reviewer for the thorough analysis of our manuscript and for valuable comments. Our responses to the Reviewer's comments and questions are written in blue font. In the manuscript, all the changes are tracked and the lines of those related to the reviewer's comments are informed in each response.

Major:

The difference between Rwork and Rfree values for structures 3CLPro-NEMO and 3CLPro C145S is still unacceptably high (see slider graphic of percentage score in the PDB report).

Since coordinates and electron density data are not released yet, this analysis is based only on the PDB validation report.

This is clearly reported in the statistics calculated in the PDB validation reports in comparison with all PDB structures as well as with comparable resolution structures. The literature references dealing with the Rwork/Rfree spread abound and I am sure are well known to all.

Protein Crystallography and Drug Discovery Jean-Michel Rondeau, Herman Schreuder, in The Practice of Medicinal Chemistry (Fourth Edition), 2015

“Since refinement programs aim at minimizing the difference between observed and calculated amplitudes (hence the R-factor), an unbiased indicator is needed to monitor the progress of refinement. Brünger proposed excluding a subset of reflections from refinement and using these reflections only for the calculation of a “free” R-factor [79]. If refinement is progressing correctly, the free R-factor will drop as well”

“The risk of overfitting the data has been considerably reduced when the concept of cross-validation was introduced [52]. This procedure involves setting aside a certain proportion of the data, the test set (ca. 5–10% of the data), which does not enter refinement, but is only taken for quality control. In this way, an independent quality indicator is generated that has a high correlation with the overall phase error and thus with the accuracy of the structural model. The free R-factor (Rfree) should remain as close as possible to the conventional crystallographic R-factor during refinement. Usually, the two values stay within 2–5% of each other if refinement proceeds well, with Rcryst having final values of 18–25% and Rfree of 22–30%. The lower the value, the better is the match between experimental data and structural model.”

In the most straightforward interpretation such a large spread between Rw and Rf indicates overfitting of the data, resulting in a poor fit of the model to the actual experimental data. Many suggestions are available in the literature on how to overcome such issues, including extensive manual rebuilding of the model, use of NCS if present, adjustment of ADP, TLS, B factors, optimization of refinement weights. Dataset/sample specific issues may affect statistics and should be discussed if statistics deviating largely from acceptable standards are reported.

Specifically for the two models:

3CLPro-NEMO solved at 2.14 Å; R_w/R_F difference is 9.5%

The R_{free} value and R_w/R_f gap indicate poor fit of the model to the data. The “slider” graphic of the PDB report shows very poor statistics for R_{free} vs all PDB structures as well as vs comparable resolution structures (actually almost the worst possible score).

Large stretches of residues are reported with poor fit to the electron density (especially chain C and D).

3CLPro C145S solved at 2.45Å; R_w/ R_f difference 7.9%. This is still high. Chain B and C show a number of residues with poor fit to the electron density. The B factors are high, this could benefit from a discussion in the manuscript.

(Here we repeat our response to Review #2 on the same point.)

We thank the reviewer for their insightful guidance regarding the crystallographic data quality. We have repeated the data processing employing some of the techniques suggested by the reviewer. Because of the triclinic space group, we had collected multiple data sets/runs with different detector distances to improve the completeness at higher resolution (wider scattering angles) and along the seams of the detector modules. We inspected the images and removed lesser quality images of some runs while paying attention to merging statistics. The weighting schemes were also adjusted during the refinements. This improved the R_{free} and R_{work} refinement statistics and the gap between them, too, while simultaneously improving the fitting of the model to the map, hence the smaller number of Ramachandran outliers. This resulted in R_{work}/R_{free} values for the C145S-Nemo structure of 30.0/24.5 compared to the previous 30.9/21.4 with 5 Ramachandran outliers, previously 6. The R_{work}/R_{free} values for the C145S structure are 19.6/23.7 with 6 Ramachandran outliers improved from 19.4/27.3 and 18 respectively. These changes have been amended in Supplementary Table S7. Details of the new refinement method used have been added to the Methods section of the manuscript (lines 913-922).

Minor:

-Figure 3b: The authors report they have removed the simulation of NEMO(190-270)-bound WT 3CLpro, in favor of the analysis of coiled-coil parameters. This is acceptable, but figure 3b needs to be removed as the manuscript contains no report on how the structure was generated and it could be misleading.

Figure 3b was removed as requested.

-Figure 3c is very difficult to understand, due to the size, lack of labeling and very succinct description in the legend. The authors should clearly indicate what values in the graphs prompts them to state “suggests a coiled-coil interruption at Tyr241, adjacent to the unfolded binding core of hNEMO,” (lines 831-832). The evidence of an “unfolded binding core of hNEMO” needs to be strengthened and more clearly explained; at this stage it would be better stated as a hypothesis.

We have increased the size of Figure 3c and added labels to improve clarity (i.e., we have added x-axis tick labels to the top panel and pointed out the region corresponding to the crystallographic NEMO peptide). The suggested coiled-coil interruption corresponds to the abrupt increase in the coiled-coil pitch observed in the plot right after Tyr241. The *putative* “unfolded binding core of hNEMO” corresponds to hNEMO₂₂₆₋₂₃₅, present in our X-ray structure in an extended conformation. With the added labels it is easier to localize the referred pitch peak. We have changed the text to better clarify these points.

-It seems like in different parts of the manuscript a reader could object that a circular argument is used to justify the computational strategies utilized (unfolded binding core of NEMO and comparative analysis of betacoronaviruses). If you allow the over-simplification

of the argument for brevity it may seem that the authors state in their rebuttal that the experimental results are validated by the computational results and the computational methods are valid because they reproduce the experimental results. If the authors could provide a more objective validation of their choice of computational methods the manuscript would be improved.

We hope we can better clarify our reasoning and logical flow of the experiments and computational analyses in this response and by making a few adjustments to the manuscript. Our experimental and computational work started from the NEMO cleavage assays, followed with coiled-coil propensity predictions, both of which helped design the NEMO constructs for the X-ray structure determination of hNEMO-3CLpro complex. We used this complex structure for the MD and QM/MM simulations of the betacoronavirus 3CLpro enzymes to predict their relative affinities towards the human NEMO peptide as a substrate.

By “unfolded binding core of NEMO”, we were referring to the hNEMO peptide present in our X-ray structure, which is found in an extended conformation. In the latest revisions, we have avoided using the wording "unfolded", and moved the section on the coiled-coil propensity analyses upstream of the X-ray structural analysis since TWISTER analysis together with the cleavage assays were useful in designing the NEMO constructs for the X-ray structure determination (Figure 1d and lines 190-202).

As for the betacoronavirus 3CLPro affinities to human NEMO peptide, we have not conducted any experiments with NEMO and 3CLpro from different betacoronaviruses. We only cite experimental studies (references 10 and 14) that demonstrate different activities of 3CLpro from SARS-CoV and SARS-CoV-2. To our knowledge, there is not yet any systematic experimental study comparing the activity of 3CLpro from different betacoronaviruses towards NEMO or other human protein substrates. Therefore, we used computational approaches to predict that and raise hypotheses about the molecular basis that could explain their relative binding affinities towards the human NEMO peptide.

-990: "the IKK α /b domains of ne NEMO homodimer: do the authors mean the IKK α /b-binding domains of NEMO?"

Thank you for pointing out the incorrect expression. Indeed we meant "IKK α / β -binding domains" where we wrote "IKK α / β domains". However, we had to correct this subclause to describe the higher-order structure proposed by Scholefield et al. (ref 39) more precisely. Now it reads "two IKK α / β domains form a head-to-head hetero tetramer thus bringing two sets of IKK α / β -NEMO complexes together (Figure 1 of Reference 39)". In doing so we no longer mention the "IKK α / β -binding domains" (lines 615-617).

-990 interacts should read interact

As stated immediately above we replaced this subclause with another, and the word "interacts" was removed (line 617).

Reviewers' Comments:

Reviewer #2:

Remarks to the Author:

The authors have addressed all our comments and the main concern about the large Rwork/Rfree gap has been resolved by the authors. The quality of the X-ray structure in particular of 3CLPro-NEMO remains relatively poor. Nevertheless, I can recommend the manuscript in the present form for publication in Nature Communications.

Minor comments:

192: The correct term is the local pitch (instead of pitch).

192 and following: The local pitch increase and the resulting increase in interhelical distance lead to an instable hydrophobic core and a more exposed single helix, supporting the idea that the 3CLpro could access the target sequence.

200-202: Sounds like the design itself was based on all analysis above. However the Twister analysis was done afterwards. So better write that the "coiled coil architecture and propensity analyses support this modelling approach"

Reviewer #3:

Remarks to the Author:

The manuscript is vastly improved in this last version.

We still have a few minor observations:

The gap between Rwork/Rfree values for the C145S/NEMO structure is now acceptable, at 24.5/30.0. This is not the result of an improvement in Rfree compared to the previous refinement reported, but rather a higher Rwork. The Rw of 24.5 is somewhat high for data at 2.1 Å resolution, compared to structures deposited with similar resolution. When looking at the Structure Validation Report it seems that chain C is the one with the worst statistics: this is one of the NEMO-bound chains and the one used for some of the analysis. How does the complex of NEMO C/ Chain C compare to NEMO B/Chain B? Could this offer an explanation for the Chain C issues?

Is it possible that the use of two datasets from the same crystal caused some of the refinement's issues? The two datasets described should be analyzed and possibly combined using BLEND to assess if they are sufficiently similar [Aller, P., Geng, T., Evans, G., Foadi, J. (2016). Applications of the BLEND Software to Crystallographic Data from Membrane Proteins. In: Moraes, I. (eds) The Next Generation in Membrane Protein Structure Determination. Advances in Experimental Medicine and Biology, vol 922. Springer, Cham. https://doi.org/10.1007/978-3-319-35072-1_9]

Otherwise the authors should provide a justification for the large Rw.

Figure 1 d: "Larger P-scores implies greater likelihood of coiled-coil": it looks like the opposite?

Figure 2 (c,d,e): the stick representation should be corrected: in some places bond order is explicitly depicted (double vs single lines) and in some others just a bold stick is used. Bold sticks without bond order are generally easier to see. The graphic software can be explicitly instructed to use such depiction.

Supplementary fig S3: "Electron density (composite OMIT map) around the 514 hNEMO226-235 peptide bound into the substrate-binding pocket of chain C (light blue surface) or 3CLpro" change "or" to "of".

Also correct the sticks representation as described above.

REVIEWERS' COMMENTS / Authors' responses in blue font

Reviewer #2 (Remarks to the Author):

The authors have addressed all our comments and the main concern about the large Rwork/Rfree gap has been resolved by the authors. The quality of the X-ray structure in particular of 3CLPro-NEMO remains relatively poor. Nevertheless, I can recommend the manuscript in the present form for publication in Nature Communications.

We once again thank the reviewer for their time and valuable help to improve this manuscript.

Minor comments:

192: The correct term is the local pitch (instead of pitch).

The appropriate section has been amended in the manuscript.

192 and following: The local pitch increase and the resulting increase in interhelical distance lead to an instable hydrophobic core and a more exposed single helix, supporting the idea that the 3CLpro could access the target sequence.

The section of the manuscript has been amended with the information provided by the reviewer (lines 350-352).

200-202: Sounds like the design itself was based on all analysis above. However the Twister analysis was done afterwards. So better write that the "coiled coil architecture and propensity analyses support this modelling approach"

Actually, we did use TWISTER and PAIRCOIL at an earlier stage of our research, and we used the coiled-coil propensity analysis results along with the peptide digestion results to design our NEMO peptides for the X-ray crystallography structure determination. In the original version of our manuscript, we had the sections and figures of the TWISTER and PAIRCOIL analyses after the x-ray structure analysis since we thought the coiled-coil propensity calculations would go well with the MD and QM/MM simulations. Following Reviewer 3#'s comment during the previous round (Round 2) of our revisions, we decided to move forward the TWISTER/PAIRCOIL analyses to upstream of the X-ray structure determination in order to make the logical progression of the experiments clearer. We hope this clarifies the logical order of the experiments.

Reviewer #3 (Remarks to the Author):

The manuscript is vastly improved in this last version.

We thank Reviewer #3 for the incisive and detailed comments and suggestions in the last round as well as the two new comments which, again, helped us tremendously to improve the refinement and to choose chains B/E rather than C/F for our detailed descriptions of the NEMO peptide bound 3CLpro structure as you see below.

We still have a few minor observations:

The gap between Rwork/Rfree values for the C145S/NEMO structure is now acceptable, at 24.5/30.0. This is not the result of an improvement in Rfree compared to the previous refinement reported, but rather a higher Rwork. The Rw of 24.5 is somewhat high for data at 2.1 Å resolution, compared to structures deposited with similar resolution. When looking at the Structure Validation Report it seems that chain C is the one with the worst statistics: this is one of the NEMO-bound chains and the one used for some of the analysis. How does the complex of NEMO C/ Chain C compare to NEMO B/Chain B? Could this offer an explanation for the Chain C issues?

To improve the Rwork/Rfree values, we used BLEND as suggested by the reviewer in their next comment. After comparing and analyzing the data using BLEND, the most agreeable part of the merged data, with good statistics, went to 2.50 Å. We used this data for MR, refinement and located the waters again. This resulted in better Rwork and Rfree values (23.2%/27.6%). This data showed overall electron density map quality improvements. The updated information is included in the manuscript Methods section as well as in Supplementary Table S7. However, the validation report still indicated that Chain C and its bound hNEMO molecule had the worst statistics and quality. We believe that this may be due to the higher B-factor for chain C (45.6 Å²) than for chain B (33.4 Å²). Consequently, we adjusted the figures and analyses to focus on chain B and its bound hNEMO molecule. Throughout the manuscript, we have drawn a distinction between the hNEMO substrate that we used during co-crystallization experiments and that is resolved in chain C (hNEMO226-235), and the hNEMO substrate resolved in chain B that we perform analyses on and compare to other structures (hNEMO226-234).

Is it possible that the use of two datasets from the same crystal caused some of the refinement's issues? The two datasets described should be analyzed and possibly combined using BLEND to assess if they are sufficiently similar [Aller, P., Geng, T., Evans, G., Foadi, J. (2016). Applications of the BLEND Software to Crystallographic Data from Membrane Proteins. In: Moraes, I. (eds) The Next Generation in Membrane Protein Structure Determination. Advances in Experimental Medicine and Biology, vol 922. Springer, Cham. https://doi.org/10.1007/978-3-319-35072-1_9]

Otherwise the authors should provide a justification for the large Rw.

We thank the referee for the suggestion to use the BLEND program. We compared and analyzed the data with BLEND. The most agreeable part of the merged data, with good statistics, went to 2.50 Å. We used this data for MR, refinement and located the waters again. This resulted in better Rwork and Rfree values (23.2%/27.6%). This data showed overall electron density map quality improvements. The updated information is included in the manuscript Methods section as well as in Supplementary Table S7.

Figure 1 d: “Larger P-scores implies greater likelihood of coiled-coil”: it looks like the opposite?

Thank you very much for pointing this out. We have corrected the sentence as: “Lower P-scores implies greater likelihood of coiled-coil”.

Figure 2 (c,d,e): the stick representation should be corrected: in some places bond order is explicitly depicted (double vs single lines) and in some others just a bold stick is used. Bold sticks without bond order are generally easier to see. The graphic software can be explicitly instructed to use such depiction.

We thank the reviewer for their suggestion and have depicted the residues as bold sticks without bond order.

Supplementary fig S3: “Electron density (composite OMIT map) around the 514 hNEMO226-235 peptide bound into the substrate-binding pocket of chain C (light blue surface) or 3CLpro” change “or” to “of”.

This has been amended in the document.

Also correct the sticks representation as described above.

The document has been amended to include bold sticks without bond order.